# Influence of Diets with Varying Essential/Nonessential Amino Acid Ratios on Mouse Lifespan

**DOI:** 10.3390/nu11061367

**Published:** 2019-06-18

**Authors:** Claudia Romano, Giovanni Corsetti, Vincenzo Flati, Evasio Pasini, Anna Picca, Riccardo Calvani, Emanuele Marzetti, Francesco Saverio Dioguardi

**Affiliations:** 1Division of Human Anatomy and Physiopathology, Department of Clinical and Experimental Sciences, University of Brescia, 25124 Brescia, Italy; cla300482@gmail.com; 2Department of Biotechnological and Applied Clinical Sciences, University of L’Aquila, 67100 L’Aquila, Italy; vincenzo.flati@univaq.it; 3Istituti Clinici Scientifici Maugeri - IRCCS Lumezzane - Cardiac Rehabilitation Division, 25065 Lumezzane (Brescia), Italy; evpasini@gmail.com; 4Fondazione Policlinico Universitario “Agostino Gemelli” IRCCS, 00168 Rome, Italy; anna.picca1@gmail.com (A.P.); riccardo.calvani@gmail.com (R.C.); emarzetti@live.com (E.M.); 5Institute of Internal Medicine and Geriatrics, Università Cattolica del Sacro Cuore, 00168 Rome, Italy; 6Department of Internal Medicine, University of Cagliari, 09042 Monserrato (Cagliari), Italy

**Keywords:** amino acids intake, essential amino acids, diet, extended lifespan, mice

## Abstract

An adequate intake of essential (EAA) and non-essential amino acids (NEAA) is crucial to preserve cell integrity and whole-body metabolism. EAA introduced with diet may be insufficient to meet the organismal needs, especially under increased physiological requirements or in pathological conditions, and may condition lifespan. We therefore examined the effects of iso-caloric and providing the same nitrogenous content diets, any diet containing different stoichiometric blends of EAA/NEAA, on mouse lifespan. Three groups of just-weaned male Balb/C mice were fed exclusively with special diets with varying EAA/NEAA ratios, ranging from 100%/0% to 0%/100%. Three additional groups of mice were fed with different diets, two based on casein as alimentary proteins, one providing the said protein, one reproducing the amino acidic composition of casein, and the third one, the control group, was fed by a standard laboratory diet. Mouse lifespan was inversely correlated with the percentage of NEAA introduced with each diet. Either limiting EAA, or exceeding NEAA, induced rapid and permanent structural modifications on muscle and adipose tissue, independently of caloric intake. These changes significantly affected food and water intake, body weight, and lifespan. Dietary intake of varying EAA/NEAA ratios induced changes in several organs and profoundly influenced murine lifespan. The balanced content of EAA provided by dietary proteins should be considered as the preferable means for “optimal” nutrition and the elevated or unbalanced intake of NEAA provided by food proteins may negatively affect the health and lifespan of mice.

## 1. Introduction

Proteins are macromolecules serving a vast array of functions within the cell, and a balanced protein synthesis and degradation is crucial for preserving cell homeostasis. Hence, increased proteolysis and reduced protein synthesis have been associated with the severe depletion of body protein reserves, eventually resulting in malnutrition. This may impact the progression of several disease-associated conditions [1]. For example, muscle wasting in people aged over 65 years old, hospitalized for a variety of chronic disease conditions, has been related with the disarrangement of protein balance [2].

Therefore, an adequate nutritional supply of protein may represent a relevant means for the management of patients who are malnourished as a consequence of reduced food intake or increased metabolic demand (i.e., chronic and acute diseases).

Dietary proteins with their various compositions of amino acids (AAs) are nitrogen (N) sources for almost all organisms. From a nutritional point of view, AAs can be classified as non-essential (NEAA) or essential (EAA), depending on their potential to be synthesized endogenously or not [3,4], although the original definitions of the two terms focused on their efficiency in promoting protein deposition [5,6]. 

Adequate dietary provision of AAs is essential for the growth, development, health, and survival of animals and humans [7]. It is also established that the administration of an adequate EAA+NEAA mix favors an increase in rat body weight, which is considered an appropriate parameter to evaluate the success of the animal in terms of growth and wellness [3,7,8,9]. However, this concept should be profoundly reconsidered if associated to lifespan. Indeed, caloric restriction and short-term caloric deficit improve the efficiency of mitochondria in humans just as in rodents, which might have the potential to increase their longevity [10].

Previous work by our group showed that the supplementation of a laboratory standard diet, containing special EAA formulations, increased rodents’ lifespan in older mice [11]. At the molecular level, such a dietary regimen was able to promote mitochondrial biogenesis and to induce organelle ultrastructural changes in the heart, skeletal muscle and adipose tissue [12,13,14]. In addition, an EAA-rich diet prevented liver damage induced by chronic ethanol consumption [15,16], boosted the effects of rosuvastatin on the kidneys [17] and accelerated wound healing in late middle-aged rats, by promoting collagen integrity [18]. Furthermore, in vitro data showed that variations in the EAA/NEAA ratio might be crucial for the fate of cancer cells via the induction of apoptosis [19]. Taken as a whole, these findings indicate that varying dietary EAA/NEAA ratios may affect cell metabolism.

While dietary proteins are the major source of AAs, the exact amounts of EAA contained in animal and vegetable proteins, introduced daily with diet, are difficult to establish. Indeed, the EAA content varies considerably depending on the source. However, any dietary protein has an EAA/NEAA ratio ≤0.9 at best. In other words, we introduce a very large amount of NEAA to meet the need for EAA and the excess of NEAA must be eliminated through complex metabolic pathways [20]. 

Many individual AAs have been tested by dietary exclusion studies in rodents, in order to demonstrate their influence on metabolism and health. For example, the restriction of methionine increases the expression of FGF21 with fall-out effects on insulin-dependent glucose uptake [21]. Other studies have been carried out in rodents with the restriction of branched-chain-AAs [22] or leucine alone [23], assessing their effects on various metabolic aspects (e.g., improving glucose tolerance or white and brown adipose tissue remodeling, respectively). The effects of caloric, protein and carbohydrate restriction on animal survival and welfare have also been studied [24,25,26]. While the restriction of individual AAs and groups of AAs has been looked at extensively in terms of metabolism and ageing, no studies have been performed specifically about EAA/NEAA ratio in the context of longevity. 

Recent data from our studies on late middle-aged animals, fed for one month with iso-caloric and iso-nitrogenous diets containing different EAA/NEAA ratios, showed significant changes in body mass and blood parameters [27]. We therefore investigated the lifelong effects on male mice of iso-caloric and iso-nitrogenous special diets containing five specific EAA/NEAA ratios, compared to a standard laboratory rodent diet.

## 2. Materials and methods

### 2.1. Animals

Three-week old outbred male Balb/C mice (Envigo, Holland) were housed in plastic cages with white wood chips for bedding, in a quiet room under controlled lighting (12 h day/night cycle) and temperature (22 ± 1°C) conditions. Animals were regularly examined by veterinary doctors for their health, the maintenance of normal daily and nocturnal behavioral activities, and for criteria of increased disease burden, according to ethics standards for animal studies. The experimental protocol was conducted in accordance with the directives of the Italian Ministry of Health and complied with ‘The National Animal Protection Guidelines’. The Ethics Committee for animal experiments of IZSLER (Brescia, Italy) (the “National Reference Centre for Animal Welfare” (http://www.izsler.it)), and the Italian Ministry of Health approved all of the procedures. 

### 2.2. Diet Composition

We used three specific diets with different EAA/NEAA ratios ranging from 100%/0% to 0%/100%, as previously described [27]. A summary of diet composition and EAA/NEAA ratios is shown in Table 1.
(1)EAA-100% diet contained exclusively EAA as the source of nitrogen [11,17,18,28]. (2)EAA-30% diet (EAA-poor diet) and thus NEAA-rich diet (70% of NEAA).(3)NEAA-100% diet contained exclusively NEAA as the source of nitrogen (thus 0% of EAA).(4)Casein-Prot diet contained only whole casein protein of highest quality.(5)Casein-AA (casein-like) diet contained free AAs equivalent to the composition of casein (EAA/NEAA = 49/51).(6)A commercial standard rodent laboratory diet (StD) (Mucedola srl, Milan, Italy) was also used.

The Casein-Prot and Casein-AA diets were used as special test diets, since no composition of AAs in StD proteins was available even for the producer, and we needed to test the safety of life-long nutrition by free AAs.

All special diets provided quantitatively equal amounts of lipids, carbohydrate and micronutrients. All diets were thus iso-caloric and provided the same amounts of nitrogen, although nitrogen content was provided by different formulations of AAs or proteins, according to those presented in Table 1. All special diets were prepared for Nutriresearch s.r.l. (Milan, Italy) by Dottori Piccioni (Milan, Italy) in accordance with AIN76-A/NIH-7 rules [29].

Animals were randomly assigned to one of the six groups. Each group was fed exclusively with a specific diet [i.e., EAA-100% diet (*n* = 30), Casein-AA diet (*n* = 30), Casein-Prot (*n* = 30), StD (*n* = 40), EAA-30% diet (*n* = 30), or NEAA-100% diet (*n* = 30)]. All animals had free access to food and water.

### 2.3. Data and Sample Collections

Body weight (BW), mean food and water consumption (g/days and mL/day, respectively) were calculated weekly in all groups. Mortality was monitored daily.

Animals from the two groups fed with NEAA-100% and EAA-30% had a mortality > 70% at the 7^th^ week, and all those still surviving were euthanized for ethical reasons linked to a drop in weight, as discussed in results. Five animals from each of the other groups were euthanized after 12 and 18 months to check their morphometric and clinical parameters. Specifically, BW and nose–tail length (body length, BL) were measured. Blood samples from the hearts and urine from the bladders were immediately collected for further analysis. Glycaemia was also measured by a glucometer in venous blood samples collected from tail veins. Subsequently, the heart, kidneys, liver, spleen, *triceps surae*, retroperitoneal white adipose tissue (rpWAT) pad and brown adipose tissue (BAT) were quickly removed and weighed [30].

### 2.4. Blood and Urine Analysis

Blood samples were collected either in tubes containing the K3-EDTA anti-coagulant for cell count analysis, or in tubes without anti-coagulants for serum separation. The blood cell count was performed with a Cell-Dyn 3700 laser-impedance cell counter (Abbott Diagnostics Division, Abbott Laboratories, IL, USA). Serum and urine levels of albumin and creatinine were assessed with an ILab Aries (Instrumentation Laboratory, Lexington, MA, USA) automatic analyzer. Serum levels of haptoglobin (Hpg) [31] and the neutrophils to lymphocytes ratio (NLR) [32,33] were also assessed as inflammatory markers. These analyses were carried out by personnel of the “Division of Laboratory Animals” of IZSLER (Brescia, Italy).

### 2.5. Statistical Analysis 

Differences between experimental groups were evaluated by one-way analysis of variance (ANOVA) followed by a Bonferroni test or Student t-test when appropriate. All analyses were performed using the Primer of Biostatistics software, with statistical significance set at *p* < 0.05. 

Spearman (r) regression values were calculated and reported where appropriate. A Mantel–Cox test (*z*) was used to test survival differences between diets [34]. 

## 3. Results 

The average survival time was markedly reduced in animals fed with the two diets where EAA were absent or deficient. Indeed, NEAA-100% diet allowed a lifespan of 44.56 ± 3.85 days, while the lifespan of EAA-30%-fed animals was slightly prolonged at 53 ± 1.97 days, on average <20%. Animals fed with Casein-Prot and Casein-AA diets survived for maximum 14 and 16 months, respectively. StD-fed animals survived for maximum 22 months. The longest survival time (25 months) was observed for the EAA-100%-fed animals (Figure 1). The percentage of EAA in the diet is correlated with survival (*r* = 0.901, *p* < 0.001).

### 3.1. Parameters Evaluation after 2 Months 

#### 3.1.1. Body Weight (BW) and Length (BL)

BW and BL are determined by the quality of nitrogen intake. For the diets poorest in EAA, NEAA-100% and EAA-30% diets, there is a rapid mortality correlating with BW loss (r = 0.92, *p* < 0.000) and also BL (r = 0.99, *p* < 0.000) when compared to all diets. A relatively modest increase in EAA provided by Casein-AA or Casein-Prot diet (which each contain about 19% more EAA than EAA-30% diet), drives a BL increase comparable to those observed in EAA-100% diet and StD. On the contrary, BW is most increased by StD and either Casein-Prot or Casein-AA diet. Instead, EAA-100% diet allows a growth in BL comparable to StD, Casein-Prot and Casein-AA. 

EAA-100% diet induced the smallest BW increase among animals fed all diets compatible with prolonged lifespan. That is, the BW of EAA-100%-fed animals, while increasing if compared with the diets poorest in EAA (NEAA-100% and EAA-30%), is significantly less increased (p < 0.001) when compared to both StD and either Casein-Prot or Casein-AA diets. 

The BW and BL of mice fed with Casein-Prot and Casein-AA diets were comparable to those of animals fed StD (see Table 2). 

#### 3.1.2. Food and Water Consumption 

The amounts of food and water consumed daily were influenced markedly by different diets. NEAA-100%-fed groups rapidly stopped growing in BL and showed a dramatically rapid BW loss similar to that of EAA-30% (Figure 2A), although the food consumption and therefore caloric intake of NEAA-100% during the first two weeks was significantly higher than any other diet except StD. With the proceeding of BW loss, and particularly in the 80% animals surviving after the sixth week, food consumption also declined, and all animals died in the following two weeks (Figure 2A,B). In NEAA-100%-fed animals, a striking difference between the grams of daily food intake, and thus calories, and BW was evident. Indeed, NEAA-100% and EAA-30%-fed animals progressively decreased in BW although their caloric intake was similar to that of EAA-100%-fed animals. Indeed, the correlation (*r*) between NEAA-100% or EAA-30% food intake and BW was –0.3 or –0.5, respectively. On the contrary, there was a higher correlation between EAA-100% food intake and BW (r = 0.9, *p* < 0.001) (Figure 2A,B). Only in the case of the NEAA-100% group, water consumption increased significantly from the first week of treatment (p < 0.001, about six-fold), and then decreased progressively to around the average water intake in StD-fed animals, before dying (Figure 2C). 

#### 3.1.3. Organ Weights

The weights of specific organs (organ weight, OW) in animals fed with NEAA-100%, EAA-30% and EAA-100% diets were significantly lower than those in animals fed with StD. The weights of the kidneys, livers, BAT and *triceps surae* of the animals fed with NEAA-100% and EAA-30% diets were significantly less than those of the EAA-100%-fed animals. The OW of animals was similar in both NEAA-100% and EAA-30% diets but, interestingly, the rpWAT was near absent in those groups (Table 2). 

#### 3.1.4. Blood and Urinary Parameters 

The two groups fed with NEAA-100% and EAA-30% diets had altered blood and urine parameters when compared to StD and EAA-100%-fed mice. This is especially evident in the reduction in blood concentration of hemoglobin and albumin, in the increase in neutrophils to lymphocytes ratio (NLR), and in the reduction of albumin and increase of creatinine in the urine. Blood and urine parameters from the EAA-100%-fed group did not differ from those of the StD-fed group, except for the Hpg value which was found to be lower (Table 3). 

### 3.2. Parameter Evaluation after 12 Months. 

#### 3.2.1. Body Weight and Length 

After one year of follow-up, BW and length were still determined by the quality of nitrogen intake. The growth of mice on Casein-Prot-based diets did not differ from that of StD-fed mice. On the contrary, the EAA-100%-fed animals showed a more modest increase in BW, which remained significantly lower than the BW of the StD, Casein-Prot and EAA-100%-fed groups (Figure 3A, Table 2).

#### 3.2.2. Food and Water Consumption

The Casein-AA, Casein-Prot and EAA-100%-fed animals showed a substantially comparable daily food intake (g/day). All those groups registered a daily food consumption that was significantly reduced compared to that of the StD-fed animals, from the 4^th^ month onwards. Water consumption (ml/day) was similar among StD, Casein-AA and Casein-Prot, whereas EAA-100%-fed mice showed a significantly increased water consumption starting from the eighth month and continuing all along the remaining follow-up (Figure 3C).

#### 3.2.3. Organ Weights

Comparable to BW, mice from the EAA-100% group showed the lowest OW except for spleen. In particular, kidney and *triceps surae* weights were significantly lower than those recorded in all other groups. The heart weights of the EAA-100%-fed animals were significantly lower than in the StD and Casein-AA groups, whereas the rpWAT weights were lower in EAA-100% than in both StD and Casein-Prot groups. Liver weight was significantly higher in the StD and Casein-Prot and Casein-AA groups than in EAA-100%. BAT weight in EAA-100% was significantly lower than the Casein-Prot group, while animals fed with Casein-AA and Casein-Prot diets had a significantly increased *triceps surae* weight, when compared to the StD group. Interestingly, the Casein-AA diet induced a significant weight decrease for rpWAT, similar to that observed in EAA-100%-fed animals. On the other hand, rpWAT weights of mice fed Casein-Prot did not differ from those of mice fed with StD (Table 1).

#### 3.2.4. Blood and Urinary Parameters 

Serum concentrations of Hpg were markedly increased (about fifty times, p < 0.001) in animals fed with the Casein-AA and Casein-Prot diets as compared to StD-fed animals. In addition, mice fed with the Casein-AA and Casein-Prot diets had increased levels of serum and lowered levels of urinary albumin and creatinine. On the other hand, EAA-100%-fed animals had normal blood parameters and urinary concentrations of albumin and creatinine, and also showed a significantly lower Hpg level even compared to the StD-fed mice (Table 3).

### 3.3. Parameter Evaluation after 18 Months. 

#### 3.3.1. Body Weight and Length. 

After 18 months, only animals fed with the StD and EAA-100% diets survived (Figure 1). EAA-100%-fed animals had lower BW than the StD-fed animals, whereas BL did not vary (Table 2). The fur appearance and spontaneous motor activity of EAA-100%-fed animals seemed to be preserved far better than in StD-fed ones. In fact, the animals fed with EAA-100% showed greater vitality than the others and often clung to the cage, keeping themselves suspended without difficulty.

#### 3.3.2. Food and Water Consumption

The food consumption of EAA-fed animals was consistently lower than in StD-fed ones (Figure 3B). On the contrary, EAA-100%-fed animals showed a progressively and significantly increased water consumption (Figure 3C).

#### 3.3.3. Organ Weights.

All OW of EAA-100%-fed animals were significantly smaller than StD-fed animals (Table 2).

#### 3.3.4. Blood and Urinary Parameters 

EAA-100%-fed mice had lower NLR, lower Hpg and higher serum albumin levels in comparison to StD-fed mice. Noticeably, urinary albumin losses were also the lowest in EAA-100%-fed animals (Table 3).

### 3.4. Parameter Evaluation after 22 Months 

After 22 months, we were able to measure only BW and BL of the few surviving animals, i.e., four animals fed with StD and nine animals fed with the EAA-100% diet (Table 2). The four StD-fed mice died a few days after these last measurements. The EAA-100% fed mice, albeit showing lower BW, appeared vital, with thick and shiny furs (Figure 1). The last mouse died at the age of 25 months and belonged to the EAA-100% group.

## 4. Discussion

The main result of our study was the observation that the lifespan of mice was affected by the quality of the AAs content in the diets. Here, we have shown that the EAA-100%-fed animals lived the longest, although they had the lowest total energy (calories) intake, and the lowest BW compared to animals fed with the other diets, while growth in BL was unaffected. These observations confirm and extend previous studies in mice whose diets were supplemented with particular EAA blends, where a prolonged lifespan paralleled improved mitochondrial biogenesis and other parameters connected with healthy aging [11,12,13].

Furthermore, we also showed that the EAA-30% diet induced a progressive BW decrease by rapid loss of muscle mass and stopped growth in BL. This was followed by precocious death. 

The diet without EAA (alias NEAA-100% diet) quickly arrested development and induced a rapid decay of animals’ health, as the availability of EAA is, for instance, the main promoter of muscle protein anabolism [35]. However, the effects of NEAA-100% and EAA-30% diets were similar, suggesting that diets providing even a relatively modestly unbalanced lowering of the EAA/NEAA ratio, in our case a diet only about <20% poorer in EAA than the Casein-Prot and Casein-AA diets (common food proteins), may trigger a severe catabolic imbalance leading to body consumption and premature death. Since the animals fed with NEAA-100% and EAA-30% diets ate less than those fed with StD, at least in the last weeks of their short lives, this could lead to the obvious conclusion that the effects were due to quantitative (and thus caloric) malnutrition. However, the daily consumption of the NEAA-100% and EAA-30% diets was comparable to the consumption of the EAA-100% diet, but in this latter case, the animals survived the longest and even longer than the StD-fed animals, albeit with reduced growth in weight (BW), but not length (BL). On this basis, we suggest that the EAA/NEAA ratio played a more prominent role than calories in ensuring animal well-being and survival. 

Special NEAA-100% and EAA-30% (thus NEAA-70%) diets strongly reduced the mass of organs and determined a complete loss of rpWAT with a proportional decrease of BAT. This was unexpected, especially for the EAA-30% diet, suggesting again that a minor reduction in the EAA/NEAA ratio (EAA were <20% lower than in the Casein diets) can lead to extremely serious consequences for the whole body. In addition, we observed that malnutrition induced by NEAA-100% and EAA-30% diets led to a decrease in serum hemoglobin and albumin values. This was probably due to reduced protein synthesis and higher turnover, dependent on a poor EAA availability. However, direct and inhibitory effects exerted by elevated plasma NEAA on albumin synthesis cannot be excluded. In fact, this would be in agreement with previous observations in adult animals [27] and in undernourished patients [36,37]. 

Significant changes, induced by NEAA-100% and EAA-30% diets, were also observed for serum Hpg levels. Hpg is a hemoglobin-binding protein synthesized in the liver and released into the circulation, where it acts as an acute phase reactant protein. In fact, it increases during acute conditions such as infection, injury, tissue destruction, some cancers, burns, surgery or trauma, in response to inflammation. On the contrary, it decreases under other pathological conditions such as chronic liver disease, hematoma and hemolytic anemia. Because Hpg levels become depleted in the presence of large amounts of free hemoglobin, a decreased Hpg is considered a good marker of hemolysis [38,39]. In our experimental setup, the NEAA-100% and EAA-30% diets induced a sharp decrease in the serum Hpg level, but also induced higher NLR, with a concomitant decrease in hemoglobin concentration, red blood cell number and spleen mass. So, we believe that the very low level of Hpg observed in NEAA and EAA-30% diets was due not only to hemolytic events, but also to an impairment of Hpg synthesis by the liver. We also observed that the EAA-100% diet reduced Hpg level, but not NLR ratio, more than StD. This confirms that EAA have anti-inflammatory activity as observed previously [11,40]. 

All special diets were consumed in significantly lower quantities than StD, thus leading to a decreasing proportion of caloric intake. It is possible that StD has a more pleasing taste than other diets. However, in our recent work, we have observed that when mice can choose between special diets and StD, the StD is not the first choice [27]. Instead, previous studies have indicated an association between increased plasma AA concentration and decreased appetite [41]. So, the quick and free AAs availability provided by special diets can trigger satiety signals, thereby decreasing food intake. Interestingly, in the case of the diets containing casein, low intake did not influence BW and OW in comparison to StD. This suggests that caloric intake is not the only parameter that influenced BW. This agree with previous works showing that EAA/NEAAA ratio plays a pivotal role in changes in body composition [3,9,27]. In addition, according to previous authors, a difference in BW and OW between EAA-100% diet and StD was observed, and we believe that this depends on the slowdown in growth caused by the EAA-100% diet.

Curiously, NEAA-100%-fed mice had an early and sharp increase in water intake without an increase in water retention. This finding agrees with previous observations in adult mice [27]. We believe that this may be related to enhanced muscle proteolysis and hyperosmolarity due to the increased release of different N-related products (such as creatinine) into the bloodstream. Indeed, all NEAA-100% and EAA-30%-fed mice had higher urinary creatinine levels compatible with muscular wasting induced by EAA deficiency. On this basis, we propose that animals fed with these diets could be used as an experimental model for muscle loss, since such a model would be inexpensive, easy to reproduce, and can be efficiently reversed by re-nutrition. 

Besides StD, in line with previous studies [29,42], as a comparison we used two diets containing, respectively, casein in the form of the whole protein (Casein-Prot) or in the form of free AAs (Casein-AA) equivalent to the composition of casein, establishing a composition of reference (see Table 1). This was suitable for the purpose of comparing a whole protein needing digestion to its free AAs composition, which was more rapidly and completely available for absorption. Casein is a widely used protein in rodent pellets, but it has been shown not to provide sufficient amounts of sulphur-containing AAs, and therefore should be integrated with other proteins from animal sources in order to match the animals’ needs for sulphur-providing AAs [29].

We compared two casein diets (-Prot and -AA) to evaluate possible biological differences between feeding proteins, which must be digested prior to absorption, and free AAs. Our choice was dictated by the fact that, unfortunately, an AAs composition of the commercially available StD was not available, since it contains 15% of unspecified “fish-based proteins” whose AAs composition is unknown to producers. On the contrary, casein-based diets were fully controlled in terms of AAs composition. Furthermore, an earlier study showed that the minimum concentration of casein which supports adequate growth, reproduction and lactation in mice was 13.6%, supplying 5.9 mg of total nitrogen/Cal [43]. Our Casein-Prot diet provided a 20% concentration of protein, thus ensuring adequate amounts of nitrogen to support growth, albeit with the cited and well known deficiencies in methionine and cisteine that identify this protein as not fully adequate to maintain a maximally prolonged life [29]. This is why we found that animals fed with Casein-Prot and Casein-AA diets had a drastically reduced survival in comparison to StD. Furthermore, Casein-Prot-fed animals lived shorter than Casein-AA-fed animals. These results might be interpreted on the basis of two main points: First, protein digestion was incomplete, since efficiency of protein digestion is <80% [44], indigested proteins contribute to fecal composition, and may thus have influenced both intestinal microbiota and promoted syntheses of some toxic metabolites [35,45]; second, the more efficient absorption of free AAs, coupled with the reduced metabolic costs of producing digestive enzymes [46] may have been responsible for survival differences. Thus, in line with previous clinical studies [47], we have shown here that the ingestion of EAA in a free form is a more efficient anabolic stimulus than the ingestion of a similar amount of AAs in the form of proteins, and that this kind of superior N intake can significantly affect life expectancy.

Animals fed with casein diets had no evident modifications of BW and BL, but lifespan was shorter compared to those on StD. However, although the animals ate significantly less calories and macronutrients than those on StD, their muscle mass increased significantly. This is in line with a previous work which has shown that casein protein intake could stimulate muscle protein synthesis without influencing lipid metabolism [48]. This observation opens some questions about the possible puzzling roles of methionine and cysteine in controlling protein syntheses and healthy life in humans and rodents. It is interesting to notice also that, by a different protocol, methionine restriction has been linked to improved lifespan in rodents, but administration of cisteine in a ratio suitable to match total sulphur containing amino acids needs, and also reducing methionine-linked toxicity, was not contemplated [20,49].

In any case, after 12 months, animals fed with casein diets had higher serum albumin (particularly in those fed with the Casein-Prot diet). High serum albumin, concomitantly with decreased urine albumin, suggests an improvement in both globular blood proteins synthesis and nephron function [50]. However, both casein diets also resulted in very high values of Hpg, suggesting that these diets provided some deficit-inducing high levels of chronic inflammation and potentially leading animals to their premature death when compared with StD or EAA-100% diet. 

Indeed, after 12 months, the urinary creatinine excretion in the Casein-Prot-fed mice was lower, whereas the blood creatinine was higher than in the StD-fed animals. This finding would suggest an impairment of kidney function [51], although urinary albumin losses were unaffected. On the contrary, in Casein-AA-fed mice, although their urinary creatinine level reached lower values than in StD-fed mice, blood creatinine did not differ from that of StD-fed mice. This would suggest a more beneficial effect on the kidneys by the free AAs intake and absorption, when compared to feeding the whole proteins. Perhaps this unidentified mechanism also provides some effects connected to a potential advantage responsible for the longer lifespan observed in Casein-AA-fed animals when compared to the Casein-Prot-fed animals.

Animals fed with the EAA-100% diet survived longer than all other groups. This is in agreement with a previous study showing that this particular EAA-blend supplementation improves mitochondrial biogenesis, thus increasing lifespan [11]. However, we also observed that these mice had slower body growth and were always (at all times) smaller than those fed with control diets. However, since no significant difference in food consumption was observed between EAA-100% and other special diets, the weight difference was not attributable to the amount of calories introduced, but was very likely due to the quality of nitrogen (thus of AAs) present in the diet. This is in agreement with a recent work where animals fed with a special blend of EAA-100% diet for one month showed similar outcomes [27]. Furthermore, other studies have demonstrated that a prolonged life is correlated with a smaller body size, both in mice [52] and in humans [53], an effect also provided by caloric restriction, which also supports our thesis.

Unexpectedly, in animals fed with EAA-100% diet, we also observed a progressive increase in water intake after eight months on this diet, although hematologic and urinary parameters did not differ from StD-fed mice. Unfortunately, to our knowledge, there is no literature that can help us to explain this behavior which, however, had no adverse effect upon or even promoted animal health. 

We also observed that the EAA-100% diet reduced inflammation, as suggested by the lower Hpg level found in animals on this diet, even when compared to StD. This is in agreement with the anti-inflammatory activity of EAA observed in previous experimental [11,40] and clinical settings [1,54,55]. Those effects on inflammation modulation could represent one of the mechanisms underpinning the longer lifespan reached by these animals.

The EAA-100% and Casein-AA diets induced a partial loss of rpWAT. rpWAT is a very plastic tissue capable of storing and releasing lipids in response to metabolic needs. A WAT decrease suggests a change in the balance of substrates used for energy production and/or an increased energy expenditure. However, it has been shown that mice fed with an L-leucine-deficient diet quickly reduced in their fat mass and lipogenic activity [56]. Our Casein-AA diet contained an adequate amount of L-leucine, whereas the EAA-Ex diet contained an even higher amount of L-leucine. Therefore, other factors besides leucine concentration are probably involved in fat loss. The reason why free AAs decreased rpWAT, and through which mechanisms, remains unclear, and further studies are necessary to have a clear picture of the mechanisms involved. In our opinion, it is the nitrogen quality in food, and not the amount of caloric intake, that determines the balance of deposited/consumed rpWAT in our study.

### 4.1. Clinical Implications

Our data suggest that it may be useful to reconsider some aspects of metabolic roles of dietary nitrogen supply in animals, as well as in humans. NEAA should be considered hidden enemies introduced through proteins, which, when introduced in excess, shorten lifespan and probably directly modulate, inhibit or blunt the synthetic activities promoted by EAA. We believe that integrating the nitrogen supply provided by diets through the supplementation of EAA, in order to increase the EAA/NEAA ratio to at least >1, should be the pivotal intervention that may most efficiently improve the life expectancy of malnourished people of any age.

The usual paradigm of clinical nutrition assumes that whatever is lacking should be provided. However, this does not seem to be true for NEAA. A better alternative would be the provision of sufficient amounts of balanced formulations of EAA, because these would better promote and maintain those metabolic pathways responsible for the synthesis of the NEAA as needed, and also of their precursors and derivatives. Indeed, balanced formulations of EAA and the unbalancing of EAA/NEAA ratios >1, promote the gene expression and activity of mTOR, PGC-1-alpha, SIRT-1, eNOS and also promote mitochondriogenesis [11]. These factors are known to be involved with optimal metabolic performances in any physiological or pathological condition and at any age, and also are effective in vitro in inducing apoptosis in cancer cells [19]. An unresolved question is now if NEAA, on the contrary, inhibit some of the pathways epigenetically activated by EAA.

### 4.2. Study limitations

Our study has some limitations that need to be discussed. We used an EAA-100% formulation tailored to human needs, and thus presently used as a nutritional supplement for humans. Furthermore, the impossibility, in our experimental settings, to conduct precise bromatological analyses of the AAs content provided by the StD, so to control the possible variations that may exist among batches even when provided by the same producer, is of some concern and causes possible unexpected bias. 

We also did not separately evaluate the contributions of individual AAs with respect to body changes. However, these probably had a mild influence on the overall results of the study. Therefore, investigating the effects of individual, or of a few, AAs, although interesting from a doctrinal point of view, would not reflect the complexity of nutritional needs and survival requirements linked to optimal animal nutrition and metabolism. This complexity is also demonstrated by testing with a known deficiency diet, such as that based only on casein protein and the peculiar amino acids ratios provided by casein.

## 5. Conclusions

We investigated the effects of different diets providing the same amount of carbohydrates, lipids, micronutrients and nitrogen, but containing various proteins or free EAA/NEAA ratios, on lifespan in mice. To our knowledge, this is the first report showing that diets providing nitrogen as free EAA are compatible with a prolonged lifespan in mice. The most relevant finding of our study was the inverse relationship between NEAA dietary content and lifespan. On the contrary, the diet with the highest amount of EAA increased lifespan and maintained low BW, while reducing systemic inflammation and preserving a balanced protein metabolism. On the contrary, the diet with a reduction of just less than 20% in EAA content compared to that usually provided by food proteins (about 45/49%) triggered the rapid catabolic processes characterized by rapid muscle mass loss, and led to precocious death of animals. We confirmed that casein, although among the reference proteins for rodents, is a protein that does not allow a lifespan comparable to that allowed by standard laboratory diets and providing adequate amounts of sulphur-containing AAs, which are insufficiently present in casein. However, in casein diets we could observe that a formulation of free AAs reproducing casein AAs content promoted the longest survival, in our opinion probably consequent to a more elevated absorption of methionine and cisteine.

It is commonly claimed that lifespan is affected by the amount of calories consumed [57]. Our data suggest that this is an oversimplified assumption. On the contrary, lifespan is conditioned by EAA and NEAA dietary content, since this ratio modulates phenotypic modifications in different organs, especially adipose and muscle tissues, and induces profound biological modifications. If we evaluate a diet in terms of the extension of lifespan, the total AAs content provided by any dietary protein should not be considered the optimal parameter. Indeed, our study suggests that an elevated intake of NEAA, provided by normal nutritional and food proteins, negatively affects health and correlates with lifespan, at least in mice. So, the concept of "optimal nutrition", being very hard to define anyway because it depends on so many variables (the aims of diet, sex, age, health, environment, species etc.), should be deeply revised on the basis of the present data. We think that the ratio among EAA and NEAA is the most likely factor responsible for the health-promoting effects of proteins in any diet, and eventually for prolonged or reduced survival, at least in rodents.

These data also led us to the question of why some AAs are provided exclusively with diet, whereas others can be synthesized by the organism itself. The origin and evolutionary significance of the relationships between EAA and NEAA is not known. We suggest as an hypothesis worth to be further explored, that the excessive introduction of NEAA may trigger certain natural selection mechanisms connected with the shortening of lifespan. 

## Figures and Tables

**Figure 1 nutrients-11-01367-f001:**
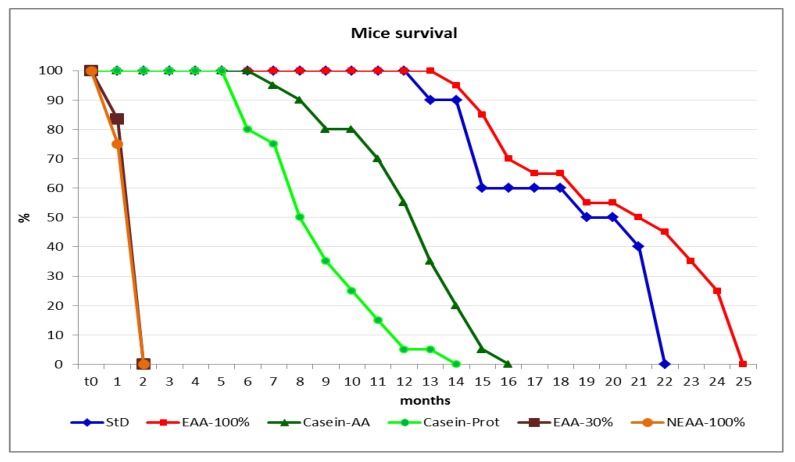
Percentage of mice’s survival according to diet. Animals fed with NEAA-100% and EAA-30% diets (orange and brown lines) had shorter lifespans when compared to StD (blue line). It is interesting to observe the different survival curves between the animals fed with the diet containing casein whole protein (light-green line) and those fed with free AAs of casein (dark-green line). In addition, note the longest survival of the mice fed with EAA-100% diet (red line). Mantel–Cox test: Casein-AA vs. Casein-Prot, z = 3.95, p < 0.001; Casein-AA vs. StD, z = 5.17, *p* < 0.001; EAA-100% vs. StD, z = 2.28, *p* = 0.0226; EAA-30% vs. NEAA-100%, z = 0.21, *p* = 0.83.

**Figure 2 nutrients-11-01367-f002:**
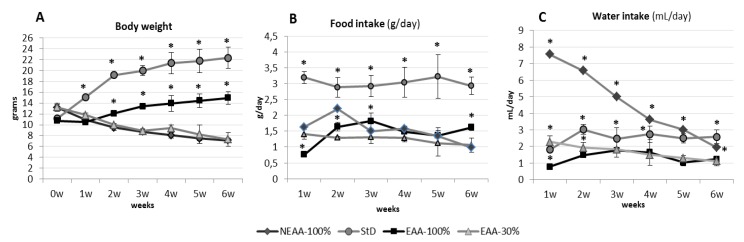
Comparison between BW (**A**), food (g/day) (**B**), and water (mL/day) consumption (**C**) (mean ± sd) of animals fed with NEAA-100% and EAA-30% diets, StD and EAA-100% diet after 6 weeks. NEAA-100% and EAA-30% diets drive rapid BW decrease, whereas EAA-100% diet slowly increases BW compared to StD (**A**). Note that EAA-100% diet was consumed in the same amount as NEAA-100% and EAA-30% diets (**B**). NEAA-100%-fed animals showed a higher water intake compared to StD, whereas EAA-30% and EAA-100%-fed animals had a lower water consumption than StD (**C**). Black square, EAA-100%; gray rhombus, NEAA-100%; gray triangle, EAA-30%; gray circle, StD. ANOVA and Bonferroni t-test, * *p* < 0.05 vs. all diets.

**Figure 3 nutrients-11-01367-f003:**
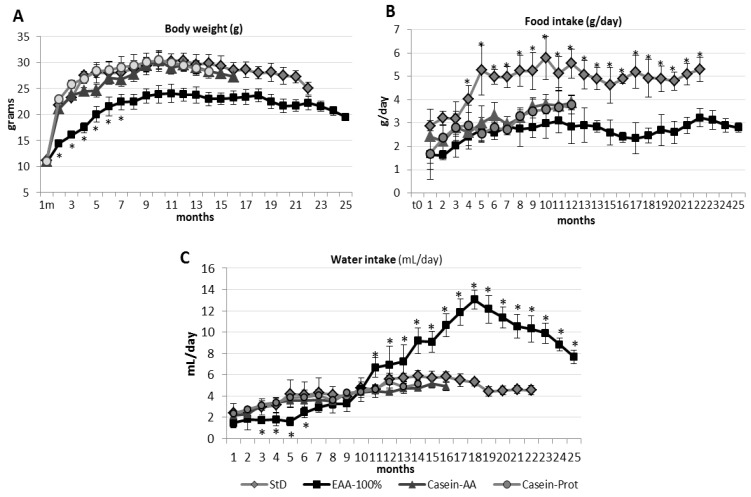
BW (**A**), food (**B**) and water (**C**) consumption (mean ± sd) according to StD, EAA-100%, Casein-Prot and Casein-AA diets during the whole survival period of mice. After about eight months, EAA-100%-fed mice significantly increase their water consumption (**C**), although food consumption (**B**) and BW (**A**) remain unchanged. See text for description. Black square, EAA-100%; gray rhombus, StD; gray triangle, Casein-AA; gray circle, Casein-Prot. ANOVA and Bonferroni t-test: * *p* < 0.05 vs. all diets.

**Table 1 nutrients-11-01367-t001:** Diet composition.

Nutrients	EAA-100%	Casein-AA	Casein-Prot	StD	EAA-30%	NEAA-100%
KCal/Kg	3995	3995	3995	3952	3995	3995
Carbohydrates %	61.76	61.76	61.76	54.61	61.76	61.76
Lipids %	6.12	6.12	6.12	7.5	6.12	6.12
Nitrogen %	20 *	20 *	20 ^	21.8 °	20 *	20 *
Proteins: % of total nitrogen content	0	0	100	95.93	0	0
Free AAs: % of total nitrogen content	100	100	0	4.07	100	100
EAA/NEAA (% in grams)	100/0	49/51	-	-	30/70	0/100
Free AAs composition (%)						
L-Leucine *(bcaa)*	31.25	9.5	--	--	9.4	--
L-Isoleucine *(bcaa)*	15.62	6	--	--	4.7	--
L-Valine *(bcaa)*	15.62	6.5	--	--	4.7	--
L-Lysine	16.25	7	--	0.97	6.24	--
L-Threonine	8.75	4	--	--	2.7	--
L-Hystidine	3.75	2.8	--	--	1.1	--
L-Phenylalanine	2.5	5	--	--	0.8	--
L-Cysteine	--	0.8	--	--	--	--
L-Cystine	3.75	--	--	0.39	1.1	--
L-Methionine	1.25	2.5	--	0.45	0.4	--
L-Tyrosine	0.75	5	--	--	2.6	1.0
L-Triptophan	0.5	1.3	--	0.28	0.01	--
L-Alanine	--	3.2	--	--	24.0	35.0
L-Glycine	--	2.4	--	0.88	10.39	15.0
L-Arginine	--	3.4	--	1.1	13.5	14.0
L-Proline	--	9.5	--	--	8.2	12.0
L-Glutamine	--	9.5	--	--	3.0	12.0
L-Serine	--	5.1	--	--	4.1	*6.0*
L-Glutamic Acid	--	9.5	--	--	2.5	2.0
L-Asparagine	--	3.5	--	--	0.79	2.0
L-Aspartic Acid	--	3.5	--	--	1.1	1.0

* Nitrogen (%) from free amino acids (AAs) only. ° Nitrogen (%) from vegetable and animal proteins and added AAs. ^ whole casein protein. EAA-100% = free essential amino acid-exclusive diet; Casein-AA = Casein-like free AAs diet; Casein-Prot = Casein whole protein diet; StD = Standard diet; EAA-30% = free essential amino acid-poor diet; NEAA-100% = non-essential free amino acid-exclusive diet. The black line represents the limit between EAA (above) and NEAA (below). *bcaa* = branched-chain amino acids.

**Table 2 nutrients-11-01367-t002:** Body weight (BW), body length (BL) and organ weight after 2, 12 and 18 months (mean ± sd).

	NEAA-100%	EAA-30%	EAA-100%	StD	Casein-Prot	Casein-AA		
2 months							*F*	*p*
Body W. (g)	7.09 ± 0.41 *^	7.31 ± 0.52 *^	14.93 ± 0.62 *	22.35 ± 1.9	22.02 ± 1.2	21.84 ± 0.9	238.26	0.000
Body L. (cm)	6.61 ± 0.1 *^	6.63 ± 0.12 *^	9.15 ± 0.13 *	9.66 ± 0.1	9.62 ± 0.2	9.47 ± 0.17	1024.88	0.000
Heart (g)	0.08 ± 0.015*	0.08 ± 0.01*	0.08 ± 0.03 *	0.13 ± 0.006	-	-	9.75	0.000
Kidneys (g)	0.12 ± 0.008*^	0.12 ± 0.009*^	0.24 ± 0.028 *	0.40 ± 0.05	-	-	105.71	0.000
Liver (g)	0.32 ± 0.049*^	0.34 ± 0.61*^	0.75 ± 0.066 *	1.10 ± 0.17	-	-	70.95	0.000
Spleen (g)	0.02 ± 0.004 *	0.02 ± 0.003 *	0.05 ± 0.008 *	0.11 ± 0.032	-	-	32.35	0.000
rpWAT (g)	0 *	0 *	0.02 ± 0.002 *	0.11 ± 0.02	-	-	123.97	0.000
BAT (g)	0.02 ± 0.01*^	0.02 ± 0.009*^	0.09 ± 0.01 *	0.13 ± 0.015	-	-	112.59	0.000
Triceps (g)	0.07 ± 0.01*^	0.06 ± 0.002*^	0.10 ± 0.011 *	0.19 ± 0.01	-	-	215.38	0.000
12 months							*F*	*p*
Body W. (g)	-	-	24.51 ± 1.9 *	31.78 ± 1.69	30.39 ± 2.79 ^	29.56 ± 1.0 ^	12.95	0.000
Body L. (cm)	-	-	9.82 ± 0.12	10.0 ± 0.13	10.01 ± 0.14	9.87 ± 0.13	2.65	0.085
Heart (g)	-	-	0.15 ± 0.02 *°	0.23 ± 0.03	0.18 ± 0.02	0.22 ± 0.05	6.51	0.000
Kidneys (g)	-	-	0.46 ± 0.06 *°§	0.8 ± 0.1	0.7 ± 0.05	0.69 ± 0.06	21.0	0.000
Liver (g)	-	-	1.36 ± 0.13 *°	1.79 ± 0.2	1.56 ± 0.2	1.7 ± 0.2	5.13	0.011
Spleen (g)	-	-	0.12 ± 0.03	0.15 ± 0.06	0.18 ± 0.02	0.15 ± 0.04	1.85	0.179
rpWAT (g)	-	-	0.07 ± 0.02 *§	0.17 ± 0.01	0.16 ± 0.08	0.09 ± 0.03 *	6.39	0.005
BAT (g)	-	-	0.13 ± 0.02 §	0.2 ± 0.04	0.23 ± 0.06	0.17 ± 0.03	5.62	0.008
Triceps (g)	-	-	0.18 ± 0.01 *°§	0.26 ± 0.02	0.31 ± 0.02 *	0.34 ± 0.03 *	54.35	0.000
18 months							*t*	*p*
Body W. (g)	-	-	23.67 ± 1.15 *	28.31 ± 1.04	-	-	6.692	0.000
Body L. (cm)	-	-	9.79 ± 0.11	9.95 ± 0.13	-	-	2.101	0.069
Heart (g)	-	-	0.15 ± 0.01 *	0.2 ± 0.01	-	-	7.906	0.000
Kidneys (g)	-	-	0.47 ± 0.04 *	0.68 ± 0.07	-	-	5.824	0.000
Liver (g)	-	-	1.18 ± 0.09 *	1.6 ± 0.15	-	-	5.369	0.000
Spleen (g)	-	-	0.08 ± 0.03 *	0.13 ± 0.04	-	-	2.236	0.000
rpWAT (g)	-	-	0.07 ± 0.03 *	0.11 ± 0.02	-	-	2.481	0.038
BAT (g)	-	-	0.12 ± 0.01 *	0.17 ± 0.01	-	-	7.906	0.000
Triceps (g)	-	-	0.15 ± 0.01 *	0.21 ± 0.01	-	-	9.487	0.000
22 months							*t*	*p*
Body W. (g)	-	-	22.2 ± 1.56 *	25.01 ± 1.2	-	-	4.515	0.000
Body L. (cm)	-	-	9.75 ± 0.12	9.81 ± 0.14	-	-	0.794	0.444

Note the similar changes in organ weight in EAA-30% and NEAA-100%-fed animals compared to StD after 2 months. Furthermore, it should be noted that the EAA-100% diet causes a slowdown in the BW and OW at all times compared to StD. ANOVA and Bonferroni t-test: * *p* < 0.05 vs. StD, ^ *p* < 0.05 vs. EAA-100%, ° *p* < 0.05 vs. Casein-AA; § *p* < 0.05 vs. Casein-Prot. WAT, white adipose tissue. BAT, brown adipose tissue.

**Table 3 nutrients-11-01367-t003:** Blood and urine data (mean ± sd) after 2, 12 and 18 months of treatment. See text for description.

	NEAA-100%	EAA-30%	EAA-100%	StD	Casein-Prot	Casein-AA		
Blood 2 months							*F*	*p*
Glucose (mg/dL)	108.3 ± 6.7 *^	111.7 ± 7.9 *	122.04 ± 8.1	127.3 ± 9.6	-	-	8.47	0.000
Erythrocytes (M/μL)	7.75 ± 1.93	8.01 ± 1.32	9.04 ± 0.58	9.13 ± 0.29	-	-	1.69	0.210
Hemoglobin (g/dL)	10.23 ± 2.37 *^	11.46 ± 2.15 *^	14.18 ± 1.16	14.8 ± 0.46	-	-	8.04	0.002
NLR	1.88 ± 1.02 *^	1.64 ± 0.93 *^	0.68 ± 0.26	0.67 ± 0.25	-	-	3.95	0.028
Albumin (g/L)	22.58 ± 1.68 *^	24.75 ± 1.56 *	26.8 ± 1.83	28.66 ± 2.57	-	-	9.03	0.000
Creatinine (μmol/L)	25.92 ± 4.54	24.95 ± 3.9	22.8 ± 1.75	23.82 ± 1.89	-	-	0.86	0.480
Haptoglobin (mg/mL)	0.02 ± 0.01 *^	0.04 ± 0.01 *^	0.13 ± 0.02 *	0.18 ± 0.02	-	-	113.83	0.000
Urine 3 months								
Albumin (g/L)	0.9 ± 0.4 *^	0.8 ± 0.6 *^	2.2 ± 0.4	1.8 ± 0.3	-	-	12.19	0.000
Creatinine (μmol/L)	5502 ± 443 *^	5653 ± 520 *^	3956 ± 824	4066 ± 1027	-	-	7.49	0.002
Blood 12 months							*F*	*p*
Glucose (mg/dL)	-	-	119.3 ± 11.4	124.3 ± 19.4	127.25 ± 14	113.25 ± 4.3	1.04	0.40
Erythrocytes (M/μL)	-	-	9.65 ± 0.62	9.93 ± 0.37	9.28 ± 0.33	9.8 ± 0.28	2.23	0.124
Hemoglobin (g/dL)	-	-	13.35 ± 1.02	14.4 ± 1.5	14.32 ± 0.3	14.4 ± 0.2	1.54	0.243
NLR	-	-	0.71 ± 0.1	0.72 ± 0.12	0.65 ± 0.07	0.67 ± 0.14	0.45	0.723
Albumin (g/L)	-	-	29.7 ± 1.7	30.88 ± 2.1	46.2 ± 8.03 *^	31.6 ± 6.6 ^	11.09	0.000
Creatinine (μmol/L)	-	-	46.9 ± 2.3	42.9 ± 5.7	72.33 ± 12.7 *^	57.0 ± 12.4	9.75	0.000
Haptoglobin (mg/mL)	-	-	0.10 ± 0.04 *	0.15 ± 0.07	4.53 ± 0.15 *^	4.97 ± 0.06 *^	3968	0.000
Urine 12 months								
Albumin (g/L)	-	-	2.13 ± 0.23	1.94 ± 0.21	1.3 ± 0.19 *^	1.02 ± 0.35 *^	21.46	0.000
Creatinine (μmol/L)	-	-	3581 ± 526	3832 ± 364	2934 ± 355 *^	2595 ± 222.7 *^	11.13	0.000
Blood 18 months							*t*	*p*
Glucose (mg/dL)	-	-	127.2 ± 8.3	133.4 ± 12.6	-	-	0.919	0.385
Erythrocytes (M/μL)	-	-	9.73 ± 0.25	9.97 ± 1.06	-	-	0.493	0.635
Hemoglobin (g/dL)	-	-	14.23 ± 0.15	14.87 ± 1.4	-	-	1.016	0.339
NLR	-	-	0.95 ± 0.2 *	1.44 ± 0.08	-	-	5.087	0.000
Albumin (g/L)	-	-	29.1 ± 0.7 *	27.4 ± 0.78	-	-	2.332	0.04
Creatinine (μmol/L)	-	-	38.3 ± 2.69	37.8 ± 1.45	-	-	0.366	0.724
Haptoglobin (mg/mL)	-	-	0.33 ± 0.15 *	0.56 ± 0.16	-	-	2.268	0.05
Urine 18 months								
Albumin (g/L)	-	-	1.58 ± 0.99 *	3.49 ± 0.87	-	-	3.241	0.01
Creatinine (μmol/L)	-	-	2954.9 ± 927	3421.7 ± 670	-	-	*0.913*	*0.388*

NLR = Neutrophils lymphocytes ratio. ANOVA and Bonferroni t-test: * *p* < 0.05 vs. StD, ^ p < 0.05 vs. EAA-100%.

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
