# Peer review of "Influence of Diets with Varying Essential/Nonessential Amino Acid Ratios on Mouse Lifespan"

_nutrients, 2019, doi:10.3390/nu11061367_

Round 1
Reviewer 1 Report
The manuscript entitled ‘The influence of diets with varying essential/nonessential amino acids ratios on mice lifespan’ evaluates the effect of different proportions of essential and non-essential amino acids on lifespan, body and tissue mass, food and water intake and some blood and urine parameters. The main conclusion of the paper is that the proportions of dietary essential and non-essential amino acids affect lifespan and is an important consideration for diets promoting human health. The authors conclude that NEAAs are detrimental to health and should be avoided.
While the topic is interesting, in my opinion, the conclusions are overstated, the results are over interpreted and the experimental design is not sufficient to answer any of the multiple questions posed. It is well known that removing an essential nutrient (also including single amino acids, e.g. leucine) leads to early death in mice. This has been shown many times.
The reference diet is not an appropriate control for any of the experimental diets. The experimental diets use either free amino acids, free amino acids from casein or whole casein protein. The reference diet contains protein from mixed sources with an undefined EAA/NEAA ratio. Furthermore, the experimental conditions are not physiological or achievable (in a human context) and therefore provide little insight into the question of optimal dietary amino acid content.
Specific comments
1) Figure 2D-F: These are unnecessary graphs. They could all be replaced with a graph showing: g food/ kg body weight/day The graph labels indicate body weight change, but the graph shows absolute body weight. In Figure 2D there is a typo. NEAAR-R.
2) There appear to be no statistics performed on the lifespan curves.
3) The level of blood hpg in the casein diets are very high, suggesting that the diet is inducing a strong systemic inflammatory response. This is surprising seeing as a majority of protein in many laboratory diets is sourced from casein. It is also very surprising that the casein diet resulted in a dramatically shortened lifespan compared to the Std diet. Are the authors sure that the casein used or the food was not contaminated?
4) Line 237-238. It is well known that limiting the supply of essential nutrients (even just one amino acid, such as leucine) leads to malnutrition and death. It is inaccurate to describe this as sarcopenia-wasting-cachexia. It is malnutrition.
5) Line 384-389 – This paragraph does not reflect the content of the manuscript. Please remove.
6) Line 373 - It is well established that calorie restriction extends lifespan. It is also well establisehd that reduced EAA intake extends lifespan. This manuscript relates to malnutrition and not the processes regulating aging.
7) the reference diet used (Std) is not an appropriate control.
Author Response
Please see attached.
Comments and Suggestions for Authors
The manuscript entitled ‘The influence of diets with varying essential/nonessential amino acids ratios on mice lifespan’ evaluates the effect of different proportions of essential and non-essential amino acids on lifespan, body and tissue mass, food and water intake and some blood and urine parameters. The main conclusion of the paper is that the proportions of dietary essential and non-essential amino acids affect lifespan and is an important consideration for diets promoting human health. The authors conclude that NEAAs are detrimental to health and should be avoided.
While the topic is interesting, in my opinion, the conclusions are overstated, the results are over interpreted and the experimental design is not sufficient to answer any of the multiple questions posed. It is well known that removing an essential nutrient (also including single amino acids, e.g. leucine) leads to early death in mice. This has been shown many times.
The reference diet is not an appropriate control for any of the experimental diets. The experimental diets use either free amino acids, free amino acids from casein or whole casein protein. The reference diet contains protein from mixed sources with an undefined EAA/NEAA ratio. Furthermore, the experimental conditions are not physiological or achievable (in a human context) and therefore provide little insight into the question of optimal dietary amino acid content.
First of all, we are very grateful to all referees for the attention to our study and data. We peculiarly thank them for observations, suggestions and helpful remarks. We have fully revised and practically re-written text in any part.
We agree with the statement that removing a single essential nutrient, as any single essential amino acid, leads to early death in mice. The use of casein, a reference proteins for mice but lacking sufficient amounts of sulphur containing amino acids (methionine and cisteine/cistine). Accordingly, we have changed, and hopefully simplified identification of the different diets, and peculiarly, we have focused attention on essential amino acids content of diets: 0%, 30%, about 55% (Standard laboratory diet, casein as whole protein and casein under form of free amino acids), 100% amino acids. This latter is a peculiarly sophisticated one, since it provides either sulphur containing amino acids providing both methionine and cistine/cisteine in a most healthy since balanced ratio according literature; Verhoef P, Steenge GR, Boelsma E, van Vliet T, Olthof MR, Katan MB. Dietary serine and cystine attenuate the homocysteine-raising effect of dietary methionine: a randomized crossover trial in humans. Am J Clin Nutr. 2004 Sep;80(3):674-9.) or tyrosine, which is an indispensable amino aicid for any organ but liver, the only organ containing metabolically significant amounts of phenyl-alanine hydroxylase necessary for synthesizing tyrosine from phenyl-alanine.
What we have shown is primarily that life is possible when rodents are fed just by free amino acids. Also, that there is a very short (near 15%) interval among essential amino acids provided by food proteins and diets providing 30% essential amino acids and a death as rapid as that provided by totally essential amino acids free ones. Please notice that the essential amino acids content of EAA30% diets, containing 20 grams of amino acids any 100 grams of pellet, provided 6 grams of essential amino acids. Most literature consider perfectly sufficient diets providing 15g of proteins (15% of pellet weight), and proteins contain approximately a maximum of 45% of essential amino acids, thus about 6.75 grams of essential amino acids any 100 grams of pellet. On the other hand, elimination of non-essential amino acids from diets not only is perfectly compatible with life but even prolongs lifespan, and this is the first evidence of this kind, at our knowledge. Although we have not tested an intermediate value among 45% of essential amino acids and 100% amino acids, a type of diet that now we know would have been interesting to be tested in a full life span, we have already tested on a shorter life span some diet providing 80% essential amino acids and 20% of selected non essential amino acids (Body Weight Loss and Tissue Wasting in Late Middle-Aged Mice on Slightly Imbalanced Essential/Non-essential Amino Acids Diet. Corsetti G, Pasini E, Romano C, Calvani R, Picca A, Marzetti E, Flati V, Dioguardi FS. Front Med (Lausanne). 2018 May 17;5:136. doi: 10.3389/fmed.2018.00136. eCollection 2018. PMID: 29868589) with results complying with those observed with 100% essential amino acids. The latter, indeed, contains cisteine/cistine ( cistine and cisteine are in continuous balance at physiological pH) and tyrosine for the said reasons, and together so providing 4.50% of non-essential amino acids, if those are considered so according a most restrictive point of view.
On those bases, we consider acceptable, if not already proven the hypothesis that more than the amount of essential amino acids, is the ratio among essential amino acids and non-essential amino acids that is indeed related to life-span. But also that primarily is of excess non-essential amino acids that is detrimental to life.
Specific comments
1) Figure 2D-F: These are unnecessary graphs. They could all be replaced with a graph showing: g food/ kg body weight/day The graph labels indicate body weight change, but the graph shows absolute body weight. In Figure 2D there is a typo. NEAAR-R.
Thank you for suggestions and help in ameliorating presentation of data and correction of our errors.
2) There appear to be no statistics performed on the lifespan curves.
We have now reported statistics on survival rate and relative p < , at IC 0.95, calculated by Mantel-Cox test.
3) The level of blood hpg in the casein diets are very high, suggesting that the diet is inducing a strong systemic inflammatory response. This is surprising seeing as a majority of protein in many laboratory diets is sourced from casein. It is also very surprising that the casein diet resulted in a dramatically shortened lifespan compared to the Std diet. Are the authors sure that the casein used or the food was not contaminated?
Yes, we are sure of the safety of our casein, since casein from the same bulk was used to prepare other pellets, and used in other experiments, and therefore of the best standard available. Indeed, referee’s note is very skilled, and linked to the protocol, since pure casein diets induce a qualitative malnutrition since lacking balanced ratios among essential amino acids, and peculiarly the sulphur-containing ones, as detailed in Reeves PG at al. AIN-93 Purified Diets for Laboratory Rodents: Final Report of the American Institute of Nutrition Ad Hoc Writing Committee on the Reformulation of the AIN-76A Rodent Diet. J. Nutr. 123: 1939-1951. (1993).
4) Line 237-238. It is well known that limiting the supply of essential nutrients (even just one amino acid, such as leucine) leads to malnutrition and death. It is inaccurate to describe this as sarcopenia-wasting-cachexia. It is malnutrition.
This is an interesting observation and worth of a peculiar semantic discussion. Indeed, we think that sarcopenia, wasting and cachexia are characterized by a sequence of peculiar biological alterations, perfectly fitting with our observations. And, yes, in our model those modifications are consequent to malnutrition. We think that both statements are correct: we describe a model of sarcopenia, wasting and cachexia dependent on qualitative ( and not quantitative) malnutrition.
5) Line 384-389 – This paragraph does not reflect the content of the manuscript. Please remove.
We have completely revised, and re-written the full paper. We hope to have met the task to be more compliant with referees advice and suggestions.
6) Line 373 - It is well established that calorie restriction extends lifespan. It is also well establisehd that reduced EAA intake extends lifespan. This manuscript relates to malnutrition and not the processes regulating aging.
Some of the authors are very discriminating about calorie restriction, we observe that it is most effective peculiarly in animals whose dimensions grow all lifelong. Efficiency of caloric restriction n primates it is still a controversial item, and as an example, calorie restriction in the elderly, indeed, is considered potentially dangerous and life threatening. Reduced essential amino acids intake to promote life span is also a questionable sentence: first of all we show that essential amino acids limitation should not be accompanied by a proportional increase in non-essential amino acids, moreover, even in Caenhorabtidis Elegans, both deficiencies and excess of different amino acids may prolong or reduce lifespan (Edwards et al. Mechanisms of amino acid-mediated lifespan extension in Caenorhabditis elegans. BMC Genetics (2015) 16:8. DOI 10.1186/s12863-015-0167-2). We have been warned in comments “that removing an essential nutrient (also including single amino acids, e.g. leucine) leads to early death in mice”, and this should sound contradictory if our study is not taken into account. We give a normal amount of nitrogen as amino acids or proteins, 20% of total food intake, although this amount is in highest percentiles of what is considered a normal intake, and it is possible that increasing those levels may be deleterious.
But, we suggest, and on the bases of the calculations of essential and non essential amino acids content of foods explicated above, that caloric restriction may be efficient, at least in part, due to a reduced total non- essential amino acids intake, because through caloric restriction non-essential amino acids intake is also reduced, and this minimize their toxicity, while body size reduction driven by caloric restriction allows survival by the amount of essential amino acids introduced with the 45% of essential amino acids provided through normal nitrogen intake . Indeed, we show that, at least in rodents, a diet based exclusively on a (well) balanced essential amino acids formulation prolongs lifespan, and this is coupled to a spontaneously reduced caloric intake and increased water intake, reduced body weight and normally developed length.
Supplementation of diets with essential amino acids have been already studied and shown to prolong lifespan better than some kind of caloric restriction (see citation 8 and 17).
On the other hand, we are well aware, as we have tested in another study, that alterations of the ratios among some essential amino acids in the formulations that we have tested is scarcely compliant with a prolonged lifespan, for instance (data not shown, will be published elsewhere).
7) the reference diet used (Std) is not an appropriate control.
We have difficulties in understanding this observation. We cannot think to a different diet than standard diet, fully compliant with what prescribed by Reeves PG at al. AIN-93 Purified Diets for Laboratory Rodents: Final Report of the American Institute of Nutrition Ad Hoc Writing Committee on the Reformulation of the AIN-76A Rodent Diet. J. Nutr. 123: 1939-1951. (1993), and that thus should be considered a normal laboratory diet suitable to be uses as standard control in a lifespan study.

Reviewer 2 Report
The manuscript “Influence of diets with varying essential/nonessential 2 amino acids ratios on mice lifespan” by Romano et al. surveys the effects of different amino acid ratios and sources in mouse diets on the longevity and overall well-being of the animals. I find the results regarding mouse lifespan very interesting, especially I the light of previous studies reporting the effects of calorie restriction on animal lifespan, and the discussion section is comprehensive and clearly presented. However, although the results are nice, the somewhat unscientific language is a little distracting every here and there. Thus, I suggest some English language editing to give the manuscript a little more polished look.
Minor comments:
In general, the reference numbers within the text are wrongly presented. Please, replace round brackets with square brackets.
Abstract:
- Are the section headlines necessary in the abstract? I suggest removal.
- The abbreviation “EAAs” is placed a bit awkwardly. Please, add “amino acids” after “essential” (“…intake of essential amino acids (EAAs)”…) or place the abbreviation with “NEAAs” in the brackets along with “respectively” (“…(EAAs and NEAAs, respectively)…”).
- Please, reconsider the keywords. At least “sarcopenia” is not relevant.
2. Materials and Methods
- Section 2.1. What are the specific characteristics of the mouse strain used in this study?
3. Results:
- Figure 1, caption: I find the style of the caption a little unscientific. Please, revise.
- Figure 2 and 3 / Tables 1 and 2, captions: The abbreviations should be defined in the caption.
4. Discussion
- Lines 287-291: How have the authors come to the conclusion that the incomplete casein digestion, fecal composition, and microbial metabolism affect the lifespan? Please, specify.
5. Conclusion
- Lines 373-378: In my opinion, there should be no references in the conclusion section. I suggest that some of this text is moved to the discussion section.
Author Response
The manuscript “Influence of diets with varying essential/nonessential 2 amino acids ratios on mice lifespan” by Romano et al. surveys the effects of different amino acid ratios and sources in mouse diets on the longevity and overall well-being of the animals. I find the results regarding mouse lifespan very interesting, especially I the light of previous studies reporting the effects of calorie restriction on animal lifespan, and the discussion section is comprehensive and clearly presented. However, although the results are nice, the somewhat unscientific language is a little distracting every here and there. Thus, I suggest some English language editing to give the manuscript a little more polished look.
We thank for the kind observations, we have fully revised text, practically re-writing most of the paper, and language has been revised by a professional editing service.
Minor comments:
In general, the reference numbers within the text are wrongly presented. Please, replace round brackets with square brackets.
Sorry for the inconvenience.
Abstract:
- Are the section headlines necessary in the abstract? I suggest removal.
- The abbreviation “EAAs” is placed a bit awkwardly. Please, add “amino acids” after “essential” (“…intake of essential amino acids (EAAs)”…) or place the abbreviation with “NEAAs” in the brackets along with “respectively” (“…(EAAs and NEAAs, respectively)…”).
- Please, reconsider the keywords. At least “sarcopenia” is not relevant.
Thank you for the suggestions and help.
2. Materials and Methods
- Section 2.1. What are the specific characteristics of the mouse strain used in this study?
We have detailed that is animals were outbread, different strains of origin, according suggestion.
Just at the purpose of informing the referee about this item, indeed, this is the second life span we performed, in a previous one, both males and females were followed lifelong, but it was an inbread population.
3. Results:
- Figure 1, caption: I find the style of the caption a little unscientific. Please, revise.
- Figure 2 and 3 / Tables 1 and 2, captions: The abbreviations should be defined in the caption.
Thank you for the suggestions and help.
4. Discussion
- Lines 287-291: How have the authors come to the conclusion that the incomplete casein digestion, fecal composition, and microbial metabolism affect the lifespan? Please, specify.
We think that there are at least two different hypotheses that can explain the different survival in this group. One is that, since protein digestion by pancreatic enzymes is always incomplete and ≤75%, thus protein indigested moieties contribute to feeding intestinal microbiota, but also to production and absorption of some toxic metabolites (g.e.: Jin U-H et al. Microbiome-Derived Tryptophan Metabolites and Their Aryl Hydrocarbon Receptor-Dependent Agonist and Antagonist Activities. Mol Pharmacol. 85: 777–788, 2014. http:// dx.doi.org/ 10.1124/mol.113.091165), while free amino acids are fully absorbed and in the upper tract of intestine ( g.e.: Goichon A et al. Enteral delivery of proteins enhances the expression of proteins involved in the cytoskeleton and protein biosynthesis in human duodenal mucosa. Am J Clin Nutr. 2015;102:359–367). Moreover, we have a full set of unpublished data on microbiome changes in time and according to quality of diets, but those findings need yet some molecular biology data dealing with intestinal barrier integrity prior to be published. Till then, our remains an hypothesis.
5. Conclusion
- Lines 373-378: In my opinion, there should be no references in the conclusion section. I suggest that some of this text is moved to the discussion section.
Thank you for the suggestion and sorry for the inconvenience. We are obsessed by carefully supporting sentences with citations

Reviewer 3 Report
General Comments
This is a potentially a well-documented and designed piece of research concerning the effect on mouse longevity of altered essential amino acids (EAAs) to NEAA ratios. However, I have several significant concerns pertaining to textual and data presentation, which must be addressed. Despite the fact I am requesting quite extensive textual changes in some cases. However, I have not request any further experimentation as the publication, from a methodological and inductive point of view, is sound.
Introduction
In general the Introduction is well-written and succinct. However, could the authors please make the following additions/adjustments:
1. 1st paragraph ‘This may impact the progression of several disease-associated conditions (1)’ Such as? Please provide some specific examples for the reader where severe depletion of the bodies protein reserves has impacted the progress of specific disorders.
2. 1st paragraph Typo ‘mean’ should be ‘means’
3. 3rd paragraph ‘Previous work from our group showed that supplementation of a laboratory standard diet (StD), containing special EAAs formulations, to old mice, for three months, increased the rodents lifespan.’ This requires a citation, please provide one or more.
4. 3rd paragraph ‘Taken as a whole, these findings indicate that varying dietary EAA/NEAA ratios may affect, and indeed somehow master, the cell and the whole-body metabolism.’ The phrase ‘somehow master’ reads very strangely. How does one ‘master’ whole body metabolism? Please re-phrase.
5. 4th paragraph ‘Moreover, the amount of EAA provided by alimentary proteins is not always adequate to match the animal needs, particularly under demanding conditions, even when the it is in good health (16).’ I am dubious about this broad claim. Moreover, you have only provided a single, somewhat obscure reference to support it. As far as I am aware of the basic literature on human protein requirements 80-100 mg/day is sufficient for a fully healthy adult individual and 20 mg/day is a survival amount without thriving. These are long established figures (see Matthews classic ‘Protein Absorption’ which summarises the knowledge up until 1991). If you are disputing that dietary protein may not be enough to supply EAAs sufficiently you need to convince me (and the knowledgeable reader) with far more evidence form the literature: for what animals do you mean and under what ‘demanding conditions’ is there evidence for? The assertion seems self-contradictory as if an animal is in ‘good health’ this assumes adequate EAA intact from the diet. As just one example, inadequate intake of dietary tryptophan or lysine via genetic mutations (both EAAs in humans) has severe outcomes, leading to Hartnup disorder and Lysinuric Protein Intolerance, respectively. See the next comment also for suggested improvements to your introduction.
6. 5th paragraph ‘. However, to our knowledge, to date there is no information available on the long-term effects of diets containing different EAA/NEAA ratios, in particular if their consumption is started immediately after weaning.’ Rather like my previous comment, be careful with this statement. There is a vast amount of literature on the effects of dietary amino acids on rodent/human health, with research conducted on all amino acids individually and by any number of categories (chemical, caloric value etc). As to your specific aim: the EAA/NEAA ratio in long-term diets and effect on longevity – you are the experts but I would still have a look at, again, Matthews from 1991 (as above) and other more recent reviews to ensure your statement does not require refinement and/or further elaboration. In particular, I think you need to at least mention that all of the individual amino acids have been tested by absence from dietary studies in rodents, other animals, and also occasionally in humans. For example, methionine restriction/absence has been looked at extensively (see PMID 28096260), branch chain amino acids, incl. leucine (e.g. PMID 29266268, 26643647), and the major uptake pathway for all the neutral EAAs (PMID 25973388). You will notice from these publications (and many others) that one of the major effects of the dietary absence/acquisition of many single/combination EAAs is metabolic re-modelling of rodents – including improved glycemic control, less body fat and induction of the endocrine hormone FGF21. There is also an extensive literature of protein dietary composition and longevity itself (see reviews PMID 28274839, 27570078, 27130207, 26718486 and others), which I am sure you are aware of. I point these things out as I think it would improve you introduction and the rationale for this paper to give more context to what is known about the dietary restriction of EAA and NEAA. At the very least it will strengthen the rationale you give: restriction of individual and groups of amino acids has been looked at in terms of metabolism and ageing extensively, but not specifically the EAA/NEAA ratio in the context of longevity.
You have introduced some of the literature on the effects of various AAs and caloric diets on rodent longevity in the discussion. Perhaps some of this could also be moved to the introduction to give the reader a better context and understanding of the current state of the field and the contribution you aim to make (see discussion comments).
Methods
Please attend to the following:
1. 2.1 Animals. ‘Just-weaned male Balb/C mice’ What day(s) were they weaned, please indicate.
2. 2.2 Diet composition. Are the percentages of EAAs and NEAAs mass or molar composition percentages? This is important to quote. Please do so here and wherever else is relevant e.g. EAA /NEAA (1:3 mass ratio) or another equivalent.
3. 2.2 Diet composition. A general point is that you need to provide more detail on the composition of your custom produced EAA-Ex, NEAA-R and NEAA-Ex diets. In particular I would like it explained how the diets were kept iso-nitrogenous? Was the nitrogen content kept constant in EAA-Ex diets by compensating for the loss of additional nitrogenous NEAAs (glutamine, asparagine) with histidine and lysine or using another non-amino acid nitrogen source? ‘Iso-nutrient’ suggests the former but it is not explained. In addition, what was the standard by which you are measuring the special diets as iso-caloric, iso-nutrient, iso-nitrogenous? Is it the standard chow diet? Please explain. I would also to see some units used to elaborate how these calculation were made. To change the amino acid composition in such dramatic fashion and keep the diets iso-caloric would imply a change in composition that would also maintain the mass of the diet, or, put another way, the mass/caloric content is constant for all diets. Is this the case? In general you need to provide some more information on how the diets were composed and how these ‘iso-‘ parameters were controlled. I see that you cannot do this for the casein-based diets (discussion) but it should be done where possible as it is a key part of this research. Can I suggest a table to document a) the composition of standard chow and then the special diets, b) listing total nitrogen content, c) caloric content per mass unit and d) NEAA/EAA ratio.
4. 2.2 Diet composition. ‘CasP and CasAA diets (Cas-based diets) were used as further control diets’. I take issue with this characterisation. Clearly based on the results the casein-based diets are not control diets – if they were, by definition, they would have the same effect as the Std diet. Please re-phrase here and wherever else in the manuscript this description is used. Rather the casein diets should be defined as alternative diets for testing your hypothesis. See my comments in Results below for more on the casein-based diet results.
5. 2.3 Sample collections. ‘Animals from the two groups fed with NEAA-based diets survived in acceptably good health for about 2 months and then they were euthanized for ethical reasons.’ This statement seems self-contradictory; how can they be in acceptably good health but still euthanised for ethical reasons? Your kill curves in Figure 1 seem to indicate many of the mice on NEAA-R and NEAA-Ex diets were dying well before this 2 month date – which, again, seems to contradict the assertion outlined in the methods. If they were dying in large percentages (as shown in Fig. 1) they could all hardly be in ‘acceptable’ good health. It is imperative that you correct these contradictions.
6. General point: please provide a list of acronyms somewhere in the manuscript. You use many and if they are located in one place it will be easier for the author to find them when required. At several points I found myself searching for what an acronym meant e.g. NLR in the tables.
Results
I have several main issues with the presentation of the results and the lack of some data which should be included.
1. ‘The main result of our study was the observation that the lifespan of mice was affected by the quality of the AAs content in the diets.’ I dislike the placement of this phrase: it is a summary conclusion not a result. Moreover, it is not specific enough or aligned with your actual research. You are looking specifically at the EAA/ENAA ratio content of different diets not the ‘quality of AAs’ generally. This phrase, or an equivalent, can be removed and re-written into the conclusions or discussion.
2. While you are at it you could also flesh out the results pertaining to the survival curves (Fig. 1), especially, as you say yourself, this is the most significant result of your research. My understanding is that often for these Kaplan-Meier curves the survival probability is quoted at different time-points. Perhaps you could quote these for the relevant time-points you have selected.
3. While you have provided good data at several time-points for mice on the different diets it is not obvious why you have provided all this data. I am assuming (based on the discussion and inductive logic) that all biometric data in Tables and figures are to give potential rationalisations for the drastic changes in mice longevity observed in Fig. 1. Fair enough, but perhaps a sentence or two making this clear in the results would be worthwhile.
4. Figure 1 legend (but also generally for all figure legends). To be honest this is a poor figure legend. Figure legends should contain all details to understand the (with the figure) the result in a self-contained manner, without recourse to the text. Your legend states conclusions, which are, in any case, stated in the discussion and conclusions already. Re-write the legend to include details required to understand the figure. You should also include the statistical analysis and abbreviation meanings in the legend.
5. Line 142. ‘NEAA-Ex and NEAA-R-fed groups rapidly stopped to grow in body length (b.l.) and showed a dramatically rapid body weight (b.w.) loss (Figure 2A).’ This is poorly worded. Body weight and length are not lost unless gained first, I think you mean they showed an attenuation or retardation of body length/weight compared with mice on the StD, correct? Please change.
6. Line 142. In the same sentence as the previous point you cite ‘Figure 2A’ for body length. This is incorrect body length is indicated in Table 1. Please fix.
7. Figure 2 key for panels A,B, and C is too small. Enlarge it to show it is the key for all 3 panels. The same change is also required for Fig. 3.
8. Why casein-based diet mice data not shown in Fig 2 and Table 1 (2 month)? These mice were alive at this stage, so why is it not shown? Please add and report inn text. If this data is not available a short explanation of why is required in either the methods or results.
9. Following on from the previous point, lines 163-165: ‘Considering that the morphometric and clinical parameters of mice fed with Cas-based diets were comparable to those StD-fed, in accordance with the guidelines for animal protection, it was considered appropriate to not sacrifice them after only 2 months.’ How can the reader determine this, you have not provided the data for casein-based diet mice for the first 2 months. As per the previous point, please add the 2 month casein-based diet mice to Fig. 2 and Table 1, or adjust the text to reflect the data that is shown.
10. Lines 170-171. ‘Both groups fed with NEAA-based diets had heavily altered blood and urine parameters if compared to StD and EAA-Ex fed mice.’ Fair enough but what are the most relevant ones, especially with reference to your later discussion? Please outline what you think are the most important results in the text.
11. Line 181. ‘CasAA-based and EAA-Ex fed animals showed a comparable daily food intake (g/day), but it was significantly lower than that of StD fed animals (Figure 3A-B).’ This is incorrect, you are only referring to data shown in Fig. 3B not Fig. 3A and B.
12. Line 184. ‘However, while growth of mice on Cas based diets did not differ from that of StD fed mice, the EAA-Ex fed animals slowly increased b.w., but it remained significantly lower than that of control groups (Table 1).’ This data is also shown in Fig. 3A, please add it.
13. Line 210. ‘Nevertheless their fur appearance and spontaneous motor activity seemed to be preserved far better than in StD-fed animals.’ How exactly did you evaluate ‘motor activity’? If you have measured it you should provide the data. If this is simply an observation you should provide a more detailed description of the symptoms and the differences between the different diets.
14. Following on from the previous point. Overall, the data presented could do with the consolidation of observational data. You have various observations scattered throughout the results and discussion. For example, line 210 and line 237 from the discussion. It would be beneficial for the reader to collect these observations into a single results section to give some coherence. You have several observations in the discussion which clearly belong in results.
15. General point: the description and use of the statistical analysis you have conducted is paltry. Please outline specifically in each figure legend and table text what test for statistical significance was used. You have simply quoted the p-values, more information is required i.e. the exact test and post-hoc test (in the case of ANOVA) and specifically what means of a normal distribution have been compared to give the specific p-value. It is not informative enough to simply have a statement in the methods.
Discussion
I normally avoid commenting too much on discussion, and yours in well written on the whole, however, there are several points to consider:
1. The first two paragraphs are largely just background information, they could be integrated into the introduction. You could move them and also elaborate your introduction as I have suggested above in my comments on the introduction.
2. Line 240. ‘However, unexpectedly, the effects of NEAA-Ex and NEAA-R diets were similar, suggesting that diets providing even a modestly unbalanced EAA/NEAA ratio, poor in EAAs, may lead to severe catabolic imbalance thus leading to body wasting and very premature death.’ Why ‘unexpectedly’? I didn’t find anything in your introduction that would suggest the severe depletion of EAAs would necessarily show any result with respect to their entire absence. Or have I missed something?
3. I think you have covered all the main points raised in the results, but as a cross-reference please ensure you have discussed the following:
· Food consumption differences – especially the wholesale reduction in consumption on all non-control (Std) diets.
· The same for body mass and organ weight.
· EAA-Ex increased water consumption from 8 months.
· Conclusions about why the CasAA and EAA-Ex diets induced a decrease in rpWAT.
· Lower value for Hpg in EAA-Ex diet mice.
Author Response
General Comments
This is a potentially a well-documented and designed piece of research concerning the effect on mouse longevity of altered essential amino acids (EAAs) to NEAA ratios. However, I have several significant concerns pertaining to textual and data presentation, which must be addressed. Despite the fact I am requesting quite extensive textual changes in some cases. However, I have not request any further experimentation as the publication, from a methodological and inductive point of view, is sound.
First of all, we are very grateful to all referees for the attention to our study and data. We peculiarly thank them for observations, suggestions and helpful remarks. We have fully revised and practically re-written text in any part.
Introduction
In general the Introduction is well-written and succinct. However, could the authors please make the following additions/adjustments:
1. 1st paragraph ‘This may impact the progression of several disease-associated conditions (1)’ Such as? Please provide some specific examples for the reader where severe depletion of the bodies protein reserves has impacted the progress of specific disorders.
2. 1st paragraph Typo ‘mean’ should be ‘means’
3. 3rd paragraph ‘Previous work from our group showed that supplementation of a laboratory standard diet (StD), containing special EAAs formulations, to old mice, for three months, increased the rodents lifespan.’ This requires a citation, please provide one or more.
4. 3rd paragraph ‘Taken as a whole, these findings indicate that varying dietary EAA/NEAA ratios may affect, and indeed somehow master, the cell and the whole-body metabolism.’ The phrase ‘somehow master’ reads very strangely. How does one ‘master’ whole body metabolism? Please re-phrase.
We have taken into account all suggestion. We thank the referee for the help and precious advices.
5. 4th paragraph ‘Moreover, the amount of EAA provided by alimentary proteins is not always adequate to match the animal needs, particularly under demanding conditions, even when the it is in good health (16).’ I am dubious about this broad claim. Moreover, you have only provided a single, somewhat obscure reference to support it. As far as I am aware of the basic literature on human protein requirements 80-100 mg/day is sufficient for a fully healthy adult individual and 20 mg/day is a survival amount without thriving. These are long established figures (see Matthews classic ‘Protein Absorption’ which summarises the knowledge up until 1991). If you are disputing that dietary protein may not be enough to supply EAAs sufficiently you need to convince me (and the knowledgeable reader) with far more evidence form the literature: for what animals do you mean and under what ‘demanding conditions’ is there evidence for? The assertion seems self-contradictory as if an animal is in ‘good health’ this assumes adequate EAA intact from the diet. As just one example, inadequate intake of dietary tryptophan or lysine via genetic mutations (both EAAs in humans) has severe outcomes, leading to Hartnup disorder and Lysinuric Protein Intolerance, respectively. See the next comment also for suggested improvements to your introduction.
This is a very skilled set of questions. We have profoundly revised text and modified acronyms, and we hope to match with the new edition most of referee’s questions. But, one point is worth of discussion since of peculiar interest, and is also one of the theoretic reason of our protocol. There is a widespread agreement that “quality” of proteins is based on essential amino acids content (g.e.: Wolfe RR et al. Factors contributing to the selection of dietary protein food sources. Clinical Nutrition 37 (2018) 130-138) , thus we should presume that requirements of proteins should match primarily essential amino acids needs, although influenced by many other factors, as total caloric intake and needs, since amino acids may be used also for energy when energetic needs are not matched by carbohydrates and lipids. That “normal” protein intake may not be adequate to match requirements in many para-physiological or pathological conditions (by us perhaps improperly defined as demanding). As an example, in aging, nitrogen requirements may not be matched by caloric food intake as calculated according usual calories per body weight ratios (g.e.: Beasley J et al. Biomarker-calibrated protein intake and bone health in the Women’s Health Initiative clinical trials and observational study. Am J Clin Nutr 2014;99:934–40. and: Ten Haff DSM et al. Insufficient protein intake is highly prevalent among physically active elderly. J Nutr Health Aging. 2018;22(9):1112-1114), and that supplementation by large amounts of essential amino acids may improve heart and muscle performances in patients affected by chronic heart failure is witnessed by different studies (Scognamiglio R et al. Impairment in Walking Capacity and Myocardial Function in the Elderly: Is There a Role for Non pharmacologic Therapy with Nutritional Amino Acid Supplements? Am J Cardiol 2008;101[suppl]:78E–81E and: , g.e.: Aquilani R et al. Despite Inflammation, Supplemented Essential Amino Acids May Improve Circulating Levels of Albumin and Haemoglobin in Patients after Hip Fractures. Nutrients 2017, 9, 637; doi:10.3390/nu9060637). On the contrary, increased protein intake may be harmful for heart health (Virtanen HEK et al. Intake of Different Dietary Proteins and Risk of Heart Failure in Men. The Kuopio Ischaemic Heart Disease Risk Factor Study. Circ Heart Fail. 2018;11:e004531. DOI: 10.1161/CIRCHEARTFAILURE.117.004531). We asked to an animal model the efficiency of essential and non-essential amino acids ratios in supporting lifespan. Therefore, we evaluated how different essential to non-essential amino acids ratios matched nitrogen intake linked metabolic needs, and although those animals were not affected by any peculiar illness, we showed that lifespan, thus health, was inversely correlated with the amount of non-essential amino acids introduced.
Obviously, the relationship among essential and non-essential amino acids intake and lifespan can be interpreted also as positively correlated with the % of total nitrogen provided as essential amino acids by different diets. Since animals had a very similar quantitative intake of aliments (daily intake of pellet was carefully measured all along the study), different qualities and not quantities of amino acids, which were the same, were determinant. This is the first study showing that since quality of nitrogen intake should be measured in terms of essential amino acids content (as widely accepted), foods providing proteins poorest in essential amino acids may be harmful not only due to an insufficient essential amino acids supply, but also because it provides amounts of non essential amino acids in elevated amounts. And we proved that regular introduction of elevated amounts of aromatic amino acids reduces lifespan.
As already written answering to the first referee, we have focused attention on essential amino acids content of diets: 0%, 30%, about 55% (Standard laboratory diet, casein as whole protein and casein under form of free amino acids), 100% amino acids. This latter is a peculiarly sophisticated one, since it provides either sulphur containing amino acids providing both methionine and cistine/cisteine in a most healthy since balanced ratio according literature; Verhoef P, Steenge GR, Boelsma E, van Vliet T, Olthof MR, Katan MB. Dietary serine and cystine attenuate the homocysteine-raising effect of dietary methionine: a randomized crossover trial in humans. Am J Clin Nutr. 2004 Sep;80(3):674-9.) or tyrosine, which is an indispensable amino aicid for any organ but liver, the only organ containing metabolically significant amounts of phenyl-alanine hydroxylase necessary for synthesizing tyrosine from phenyl-alanine.
What we have shown is primarily that life is possible when rodents are fed just by free amino acids. Also, that there is a very short (near 15%) interval among essential amino acids provided by food proteins and diets providing 30% essential amino acids and a death as rapid as that provided by totally essential amino acids free ones. Please notice that the essential amino acids content of EAA30% diets, containing 20 grams of amino acids any 100 grams of pellet, provided 6 grams of essential amino acids. Most literature consider perfectly sufficient diets providing 15g of proteins (15% of pellet weight), and proteins contain approximately a maximum of 45% of essential amino acids, thus about 6.75 grams of essential amino acids any 100 grams of pellet. On the other hand, elimination of non-essential amino acids from diets not only is perfectly compatible with life but even prolongs lifespan, and this is the first evidence of this kind, at our knowledge. Although we have not tested an intermediate value among 45% of essential amino acids and 100% amino acids, a type of diet that now we know would have been interesting to be tested in a full life span, we have already tested on a shorter life span some diet providing 80% essential amino acids and 20% of selected non essential amino acids (Body Weight Loss and Tissue Wasting in Late Middle-Aged Mice on Slightly Imbalanced Essential/Non-essential Amino Acids Diet. Corsetti G, Pasini E, Romano C, Calvani R, Picca A, Marzetti E, Flati V, Dioguardi FS. Front Med (Lausanne). 2018 May 17;5:136. doi: 10.3389/fmed.2018.00136. eCollection 2018. PMID: 29868589) with results complying with those observed with 100% essential amino acids. The latter, indeed, contains cisteine/cistine ( cistine and cisteine are in continuous balance at physiological pH) and tyrosine for the said reasons, and together so providing 4.50% of non-essential amino acids, if those are considered so according a most restrictive point of view.
On those bases, we consider acceptable, if not already proven the hypothesis that more than the amount of essential amino acids, is the ratio among essential amino acids and non-essential amino acids that is indeed related to life-span. But also that primarily is of excess non-essential amino acids that is detrimental to life.
6. 5th paragraph ‘. However, to our knowledge, to date there is no information available on the long-term effects of diets containing different EAA/NEAA ratios, in particular if their consumption is started immediately after weaning.’ Rather like my previous comment, be careful with this statement. There is a vast amount of literature on the effects of dietary amino acids on rodent/human health, with research conducted on all amino acids individually and by any number of categories (chemical, caloric value etc).
The statement is correct, there is a vast amount of literature on the effects of dietary amino acids on rodent/human health, with research conducted on all amino acids individually and by any number of categories but studies, but extremely few compared different qualities and not quantities of complex mixtures of amino acids in mammals. Most studies dal with very short lived animals and are questionable (g.e.: Ja WW et al. Water- and nutrient-dependent effects of dietary restriction on Drosophila lifespan.
www.pnas.org/cgi/doi:10.1073/pnas.0908016106).
As to your specific aim: the EAA/NEAA ratio in long-term diets and effect on longevity – you are the experts but I would still have a look at, again, Matthews from 1991 (as above) and other more recent reviews to ensure your statement does not require refinement and/or further elaboration. In particular, I think you need to at least mention that all of the individual amino acids have been tested by absence from dietary studies in rodents, other animals, and also occasionally in humans. For example, methionine restriction/absence has been looked at extensively (see PMID 28096260), branch chain amino acids, incl. leucine (e.g. PMID 29266268, 26643647), and the major uptake pathway for all the neutral EAAs (PMID 25973388). You will notice from these publications (and many others) that one of the major effects of the dietary absence/acquisition of many single/combination EAAs is metabolic re-modelling of rodents – including improved glycemic control, less body fat and induction of the endocrine hormone FGF21. There is also an extensive literature of protein dietary composition and longevity itself (see reviews PMID 28274839, 27570078, 27130207, 26718486 and others), which I am sure you are aware of. I point these things out as I think it would improve you introduction and the rationale for this paper to give more context to what is known about the dietary restriction of EAA and NEAA. At the very least it will strengthen the rationale you give: restriction of individual and groups of amino acids has been looked at in terms of metabolism and ageing extensively, but not specifically the EAA/NEAA ratio in the context of longevity.
We have peculiarly appreciated those questions. Again, facing any aspect of nutrition and longevity would be worth a full editorial. We will detail some aspect of methionine restriction and longevity, since deeply entangled with our choice of testing casein and a casein reproducing formulation of free amino acids.
Casein is deficient in methionine, this is why should not be the sole source of protein in mice pellets, and this deficiency does not improves lifespan. We have just confirmed previous knowledge and this support efficacy of our protocol, but our main goal was to test protein versus free amino acids. . Therefore, diets predicted to increase lifespan by reducing methionine may have a positive outcome for reasons that may be different by simple methionine restriction, and in our opinion the reasons should be found in the peculiar links among methionine, cisteine/cistine, methyl groups and folates metabolism.
It should be remembered that in earliest studies defining essential amino acids and non essential ones, methionine was found letal and causing anemia at doses very near to minimal dosages allowing life (Cohen HP and Berg CP. Erythrocyte turnover in rats fed diets high in methionine.J. Biol. Chem. 1956, 222:85-88. ) for reasons still unknown ten years later (Mengel CE and Klavis JV. Development of hemolytic anemia in rats fed methionine. J Nutr. 1967 May;92(1):104-110. ), and poorly understood even more recently (Sanchez-Roman I et al. Effects of aging and methionine restriction applied at old age on ROS generation and oxidative damage in rat liver mitochondria. Biogerontology.(2012).13: 399-411. DOI 10.1007/s10522-012-9384-5). The problem is that fulfilling sulphur amino acids needs just by methionine, which largely has to be transformed in cistine to cope with metabolic needs of this family of amino acid, would cause a depletion of methyl group in position 5 of folates, if sulphur amino acids requirements would not be matched by clustering sulphur amino acids family with sufficient amount of cisteine/cistine, according to a cascade of events described in any biochemistry book. Thus, minimizing methionine in models of programmed nutrition were different amounts of methionine (and only methionine) are compared, would reduce efficiency of protein syntheses and connected epigenetic modifications, in turn these reduce growth and development, and methionine toxicity, prolonging life span in the peculiar experimental setting comparing toxic levels of methionine and reduced ones. In literature there are also plenty of data showing that consequences of deficiencies in food methionine content should be matched by increased introduction to maintain growth and survival (Li H et al. Changes in plasma amino acid profiles, growth performance and intestinal antioxidant capacity of piglets following increased consumption of methionine as its hydroxy analogue. British Journal of Nutrition (2014), 112, 855–867 doi:10.1017/S000711451400172X) and there is no conundrum, just the demonstration that there is just a very narrow line among sufficient and insufficient, toxic and not toxic amounts of methionine in nutrition, and that it is theoretically misleading the study of extreme modifications of dosages of a single amino acid when it is just one component out of many components of a “functional” cluster, in this case the sulphur providing amino acids.
About FGF21 we have many data obtained from other protocols. Indeed its effects are strictly linked to the presence on cells and in plasma of a peculiar receptor, beta-Klotho, which expressed and secreted by many cells and tissues, but not all cells of all tissues. Indeed, we have shown that human heart is also a producer of, not only dependent on, beta-Klotho molecules (Corsetti G et al. Decreased expression of Klotho in cardiac atria biopsy samples from patients at higher risk of atherosclerotic cardiovascular disease. Journal of Geriatric Cardiology (2016) 13: 701-711) Liver is the main (but not unique) organ ruling circulating levels either of FGF21 and of beta-Klotho, and we may confidentially confirm that in liver and kidneys expression of both molecules is deeply influenced by EAA supplementation, although those dosages has been yet performed in livers and kidneys of both middle aged males and females of a preliminary unpublished study performed with inbread “ Swiss Type” Balb/C mice.
You have introduced some of the literature on the effects of various AAs and caloric diets on rodent longevity in the discussion. Perhaps some of this could also be moved to the introduction to give the reader a better context and understanding of the current state of the field and the contribution you aim to make (see discussion comments).
Methods
Please attend to the following:
1) 2.1 Animals. ‘Just-weaned male Balb/C mice’ What day(s) were they weaned, please indicate.
We have indicate age weeks.
2) 2.2 Diet composition. Are the percentages of EAAs and NEAAs mass or molar composition percentages? This is important to quote. Please do so here and wherever else is relevant e.g. EAA /NEAA (1:3 mass ratio) or another equivalent.
The relative weights in grams are presented in Tab.1.
3) 2.2 Diet composition. A general point is that you need to provide more detail on the composition of your custom produced EAA-Ex, NEAA-R and NEAA-Ex diets. In particular I would like it explained how the diets were kept iso-nitrogenous? Was the nitrogen content kept constant in EAA-Ex diets by compensating for the loss of additional nitrogenous NEAAs (glutamine, asparagine) with histidine and lysine or using another non-amino acid nitrogen source? ‘Iso-nutrient’ suggests the former but it is not explained. In addition, what was the standard by which you are measuring the special diets as iso-caloric, iso-nutrient, iso-nitrogenous? Is it the standard chow diet? Please explain. I would also to see some units used to elaborate how these calculation were made. To change the amino acid composition in such dramatic fashion and keep the diets iso-caloric would imply a change in composition that would also maintain the mass of the diet, or, put another way, the mass/caloric content is constant for all diets. Is this the case? In general you need to provide some more information on how the diets were composed and how these ‘iso-‘ parameters were controlled. I see that you cannot do this for the casein-based diets (discussion) but it should be done where possible as it is a key part of this research. Can I suggest a table to document a) the composition of standard chow and then the special diets, b) listing total nitrogen content, c) caloric content per mass unit and d) NEAA/EAA ratio.
We trust on a very skilled and specialized producer of custom pellets since the fifties of last century for different animals (Laboratorio Dottori Piccioni, via Guglielmo Marconi 29, Gessate, +390295380285), which prepared for us an aproteic diet providing carbohydrates, lipids and micronutrients according NIH7A rules, and the same formulation was completed by 20% (in weight) of casein, and of the four different formulations of essential amino acids that we provided already blended. Iso-nitrogenous pretended to underline that the amount of nitrogen contained provided by any diet was as much as possible similar.
Iso-caloric was used to underline than any macro-nutrient and micronutrient was identical from the beginning and not modified by quality of nitrogen introduced in the aproteic formulation.
4. 2.2 Diet composition. ‘CasP and CasAA diets (Cas-based diets) were used as further control diets’. I take issue with this characterisation. Clearly based on the results the casein-based diets are not control diets – if they were, by definition, they would have the same effect as the Std diet. Please re-phrase here and wherever else in the manuscript this description is used. Rather the casein diets should be defined as alternative diets for testing your hypothesis. See my comments in Results below for more on the casein-based diet results.
Very helpful suggestion and helpful critique. We have deeply revised text and semantics.
5. 2.3 Sample collections. ‘Animals from the two groups fed with NEAA-based diets survived in acceptably good health for about 2 months and then they were euthanized for ethical reasons.’ This statement seems self-contradictory; how can they be in acceptably good health but still euthanised for ethical reasons? Your kill curves in Figure 1 seem to indicate many of the mice on NEAA-R and NEAA-Ex diets were dying well before this 2 month date – which, again, seems to contradict the assertion outlined in the methods. If they were dying in large percentages (as shown in Fig. 1) they could all hardly be in ‘acceptable’ good health. It is imperative that you correct these contradictions.
Thank you for suggestions. we would follow all of them, and ameliorate our paper.
6. General point: please provide a list of acronyms somewhere in the manuscript. You use many and if they are located in one place it will be easier for the author to find them when required. At several points I found myself searching for what an acronym meant e.g. NLR in the tables.
We should have thought to this without suggestion. Very sorry for having omitted this polite procedure. Again, thank you for suggestions.
Results
I have several main issues with the presentation of the results and the lack of some data which should be included.
1. ‘The main result of our study was the observation that the lifespan of mice was affected by the quality of the AAs content in the diets.’ I dislike the placement of this phrase: it is a summary conclusion not a result. Moreover, it is not specific enough or aligned with your actual research. You are looking specifically at the EAA/ENAA ratio content of different diets not the ‘quality of AAs’ generally. This phrase, or an equivalent, can be removed and re-written into the conclusions or discussion.
2. While you are at it you could also flesh out the results pertaining to the survival curves (Fig. 1), especially, as you say yourself, this is the most significant result of your research. My understanding is that often for these Kaplan-Meier curves the survival probability is quoted at different time-points. Perhaps you could quote these for the relevant time-points you have selected.
3. While you have provided good data at several time-points for mice on the different diets it is not obvious why you have provided all this data. I am assuming (based on the discussion and inductive logic) that all biometric data in Tables and figures are to give potential rationalisations for the drastic changes in mice longevity observed in Fig. 1. Fair enough, but perhaps a sentence or two making this clear in the results would be worthwhile.
4. Figure 1 legend (but also generally for all figure legends). To be honest this is a poor figure legend. Figure legends should contain all details to understand the (with the figure) the result in a self-contained manner, without recourse to the text. Your legend states conclusions, which are, in any case, stated in the discussion and conclusions already. Re-write the legend to include details required to understand the figure. You should also include the statistical analysis and abbreviation meanings in the legend.
5. Line 142. ‘NEAA-Ex and NEAA-R-fed groups rapidly stopped to grow in body length (b.l.) and showed a dramatically rapid body weight (b.w.) loss (Figure 2A).’ This is poorly worded. Body weight and length are not lost unless gained first, I think you mean they showed an attenuation or retardation of body length/weight compared with mice on the StD, correct? Please change.
6. Line 142. In the same sentence as the previous point you cite ‘Figure 2A’ for body length. This is incorrect body length is indicated in Table 1. Please fix.
7. Figure 2 key for panels A,B, and C is too small. Enlarge it to show it is the key for all 3 panels. The same change is also required for Fig. 3.
8. Why casein-based diet mice data not shown in Fig 2 and Table 1 (2 month)? These mice were alive at this stage, so why is it not shown? Please add and report inn text. If this data is not available a short explanation of why is required in either the methods or results.
9. Following on from the previous point, lines 163-165: ‘Considering that the morphometric and clinical parameters of mice fed with Cas-based diets were comparable to those StD-fed, in accordance with the guidelines for animal protection, it was considered appropriate to not sacrifice them after only 2 months.’ How can the reader determine this, you have not provided the data for casein-based diet mice for the first 2 months. As per the previous point, please add the 2 month casein-based diet mice to Fig. 2 and Table 1, or adjust the text to reflect the data that is shown.
10. Lines 170-171. ‘Both groups fed with NEAA-based diets had heavily altered blood and urine parameters if compared to StD and EAA-Ex fed mice.’ Fair enough but what are the most relevant ones, especially with reference to your later discussion? Please outline what you think are the most important results in the text.
11. Line 181. ‘CasAA-based and EAA-Ex fed animals showed a comparable daily food intake (g/day), but it was significantly lower than that of StD fed animals (Figure 3A-B).’ This is incorrect, you are only referring to data shown in Fig. 3B not Fig. 3A and B.
12. Line 184. ‘However, while growth of mice on Cas based diets did not differ from that of StD fed mice, the EAA-Ex fed animals slowly increased b.w., but it remained significantly lower than that of control groups (Table 1).’ This data is also shown in Fig. 3A, please add it.
13. Line 210. ‘Nevertheless their fur appearance and spontaneous motor activity seemed to be preserved far better than in StD-fed animals.’ How exactly did you evaluate ‘motor activity’? If you have measured it you should provide the data. If this is simply an observation you should provide a more detailed description of the symptoms and the differences between the different diets.
14. Following on from the previous point. Overall, the data presented could do with the consolidation of observational data. You have various observations scattered throughout the results and discussion. For example, line 210 and line 237 from the discussion. It would be beneficial for the reader to collect these observations into a single results section to give some coherence. You have several observations in the discussion which clearly belong in results.
15. General point: the description and use of the statistical analysis you have conducted is paltry. Please outline specifically in each figure legend and table text what test for statistical significance was used. You have simply quoted the p-values, more information is required i.e. the exact test and post-hoc test (in the case of ANOVA) and specifically what means of a normal distribution have been compared to give the specific p-value. It is not informative enough to simply have a statement in the methods.
We have tied to follow all suggestions and hope to have fulfilled all the needed.
Discussion
I normally avoid commenting too much on discussion, and yours in well written on the whole, however, there are several points to consider:
1. The first two paragraphs are largely just background information, they could be integrated into the introduction. You could move them and also elaborate your introduction as I have suggested above in my comments on the introduction.
2. Line 240. ‘However, unexpectedly, the effects of NEAA-Ex and NEAA-R diets were similar, suggesting that diets providing even a modestly unbalanced EAA/NEAA ratio, poor in EAAs, may lead to severe catabolic imbalance thus leading to body wasting and very premature death.’ Why ‘unexpectedly’? I didn’t find anything in your introduction that would suggest the severe depletion of EAAs would necessarily show any result with respect to their entire absence. Or have I missed something?
3. I think you have covered all the main points raised in the results, but as a cross-reference please ensure you have discussed the following:
· Food consumption differences – especially the wholesale reduction in consumption on all non-control (Std) diets.
· The same for body mass and organ weight.
· EAA-Ex increased water consumption from 8 months.
· Conclusions about why the CasAA and EAA-Ex diets induced a decrease in rpWAT.
· Lower value for Hpg in EAA-Ex diet mice.
Thank you for the careful reading and for any suggestion. As said, we have totally revised any part of the paper, practically re-written it, and also modified acronyms to better drive also a reader less skilled than reviewer to a better understanding of our data. We hope to have ameliorate and answered most, if not all questions.

Round 2
Reviewer 1 Report
The authors have now sufficiently softened the strength of their conclusions to more accurately reflect the experiments performed. I would now consider it acceptable for publication.